# Sustainable Energy Production in Smart Cities

**Ramiz Salama** [1,*] **and Fadi Al-Turjman** [2,3]

1  Department of Computer Engineering, AI and Robotics Institute, International Research Center for AI and IoT, Near East University, Mersin 10, Nicosia 99138, Turkey
2  Artificial Intelligence Engineering Department, AI and Robotics Institute, International Research Center for AI and IoT, Near East University, Mersin 10, Nicosia 99138, Turkey; fadi.alturjman@neu.edu.tr
3  Research Center for AI and IoT, Faculty of Engineering, University of Kyrenia, Mersin 10, Kyrenia 99320, Turkey
*  Correspondence: ramiz.salama@neu.edu.tr

**Abstract:** Finding a method to provide the installed Internet of Things (IoT) nodes with energy that is both ubiquitous and long-lasting is crucial for ensuring continuous smart city optimization. These and other problems have impeded new research into energy harvesting. After the COVID-19 pandemic and the lockdown that all but ended daily activity in many countries, the ability of human remote connections to enforce social distancing became crucial. Since they lay the groundwork for surviving a lockdown, Internet of Things (IoT) devices are once again widely recognised as crucial elements of smart cities. The recommended solution of energy collection would enable IoT hubs to search for self-sustaining energy from ecologically large sources. The bulk of urban energy sources that could be used were examined in this work, according to descriptions made by researchers in the literature. Given the abundance of free resources in the city covered in this research, we have also suggested that energy sources can be application-specific. This implies that energy needs for various IoT devices or wireless sensor networks (WSNs) for smart city automation should be searched for near those needs. One of the important smart, ecological and energy-harvesting subjects that has evolved as a result of the advancement of intelligent urban computing is intelligent cities and societies. Collecting and exchanging Internet of Things (IoT) gadgets and smart applications that improve people's quality of life is the main goal of a sustainable smart city. Energy harvesting management, a key element of sustainable urban computing, is hampered by the exponential rise of Internet of Things (IoT) sensors, smart apps, and complicated populations. These challenges include the requirement to lower the associated elements of energy consumption, power conservation, and waste management for the environment. However, the idea of energy-harvesting management for sustainable urban computing is currently expanding at an exponential rate and requires attention due to regulatory and economic constraints. This study investigates a variety of green energy-collecting techniques in relation to edge-based intelligent urban computing's smart applications for sustainable and smart cities. The four categories of energy-harvesting strategies currently in use are smart grids, smart environmental systems, smart transportation systems, and smart cities. In terms of developed algorithms, evaluation criteria, and evaluation environments, this review's objective is to discuss the technical features of energy-harvesting management systems for environmentally friendly urban computing. For sustainable smart cities, which specifically contribute to increasing the energy consumption of smart applications and human life in complex and metropolitan areas, it is crucial from a technical perspective to examine existing barriers and unexplored research trajectories in energy harvesting and waste management.

**Keywords:** energy harvesting; Internet of Things; sustainable smart cities

## 1. Introduction

Energy harvesting is the process of transforming energy that would otherwise be squandered into energy that might be used to power autonomous devices (EH). Energy harvesting offers a solution to the current energy problem facing Internet of Things (IoT)

networks by enabling the physical or chemical extraction of ambient energy from natural or artificial environmental sources. On the other hand, the idea of a "smart city" refers to the level of urbanization and technological advancement in every given metropolis. Intelligence is defined differently depending on the country or city. Many definitions of this idea include the particular application domains that smart city developers are interested in as well as people's daily desires to relate to objects or oneself. Knowledge may be crucial for surviving severe calamities and crises in one city, such as earthquakes, floods, and roaring fires. Someone else may see it as a security and terrorism concern. To one person, the desire to enhance people's lives in general may be the driving force for interest in smart cities. The capacity of a smart city to connect people and objects via the internet is one of its fundamental features. For instance, there are relationships and connections between things like homespun appliances, vehicles, and buildings. This gives a city's planners access to data on the smart city, and a city often automates difficult issues. Instantaneous health care delivery services, enhanced energy and transportation networks (despite increasing urban congestion), enhanced daily human habitation, safekeeping security intelligence services, quick rescue operations in the event of a fire or natural disaster, and the development of green energy sources to reduce carbon emissions and combat climate change are just a few application areas for these concerns.

Web technology is used in the smart city to control and access physical assets. They are connected to the internet in order to make things sentient and enable interactions between them, whether they are human-to-human or machine-to-machine interactions. To provide the planner the chance to finish the positive details relating to the precise needs of the city, the smart city relies on tactile data from IoT [1,2]. This is the central idea of the smart city enabled by IoT mentality.

A network of things or people interacting with things that are hosted in the cloud is all that the Internet of Things, or IoT, is. By utilizing the different sensors found throughout the city, the Internet of Things (IoT) sends data between physically connected IoT devices, facilitating online or cloud-based communication between things or between people and objects. Because of this, the Internet of Things frequently consists of wired or wireless internet-enabled devices, such as actuators, mobile phones, telemedicine networks, sensors, RFID tags, and so on. The Internet of Things (IoT) facilitates the rapid availability of information about intelligent cities for independent planning and the ability of such cities to function for daily living and ongoing human life. This is seen in Figure 1.

Smart cities are a cutting-edge concept for managing urban regions that will improve sustainability and quality of life for residents. Moreover, in order to improve ecological and economic sustainability, projects integrating digitalization and smart cities must generate value. However, by first deducting the rewards from the efforts, this increased value can be presented more plainly.

The challenge of building their infrastructure with modern methods that utilize little energy and have little impact on the environment is one that smart cities must face. Fighting climate change and other environmental problems requires the development of "smart buildings" and a more effective transportation system. A self-managing automated system that can transform electric power into a final product with little human involvement is necessary as part of a smart city's balanced energy exchange.

In order to balance power output and consumption, reduce generation capacity, and have an impact on other energy market participants, smart cities are developing a unified system that combines diverse energy, heat, gas, and water systems as well as telecommunications structures. The long-term health of the energy industry depends on electrification, which is the process of moving civilization toward using electricity as its primary energy source.

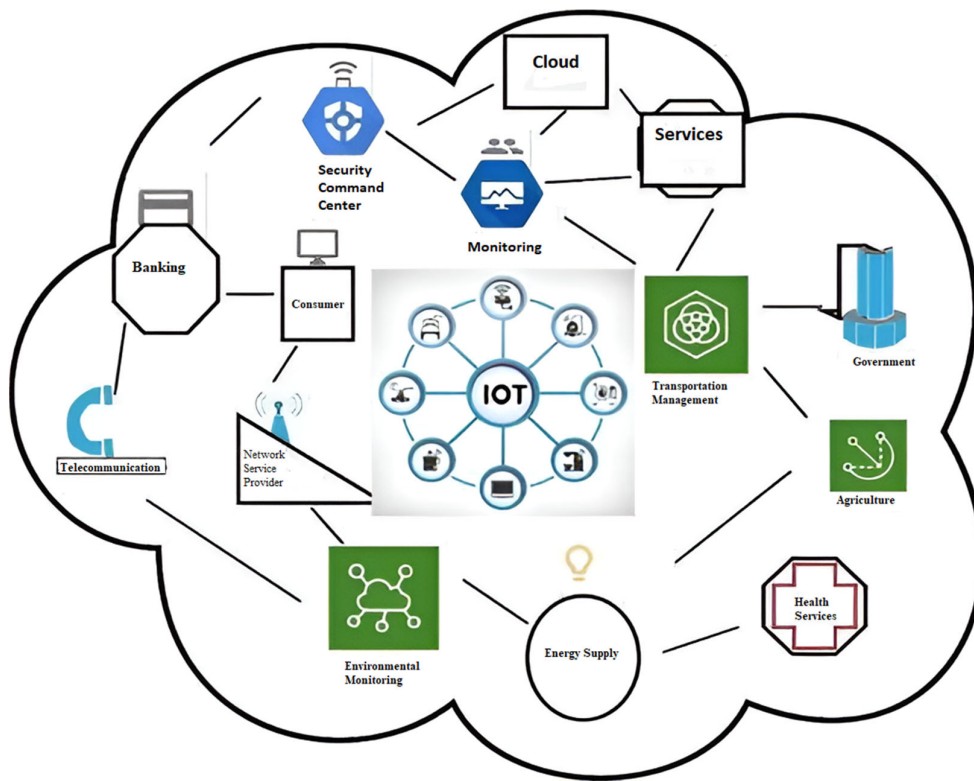

**Figure 1.** The Internet of Things.

People from all demographic groups no longer view the Internet of Things (IoT) as an impossibility. For instance, the COVID-19 pandemic and the accompanying activity lockdowns in virtually every country have made it imperative that the Internet of Things (IoT) concept be upgraded. The majority of countries have opted to use online and IoT innovation as one crucial safety net in order to maintain normalcy throughout this pandemic. The IoT's energy supply is necessary to ensure that connected devices are always functional. The problem that causes the most serious concerns regarding the adoption of the smart city and its sustainability is how IoT nodes and/or linked devices may be powered in order to keep providing smart city planners with uninterrupted data. Researchers have conducted numerous studies focusing on managing and maintaining the energy supply to first WSNs and eventually IoT nodes at various periods in time. Energy management systems can be used to lower the amount of energy used by smart buildings, according to a number of international research projects. The EU parliament published recommendations in 2002 to increase a building's energy efficiency. Eventually, according to experts, by 2022, there will be more than 500 connected smart devices in smart buildings. Therefore, it is essential to spread knowledge of the rising energy requirements of Internet of Things technology. The most important node in a smart city is the Internet of Things, which comprises gadgets that use more energy than WSNs. The current paradigm shift from concerns about energy supply to those regarding Internet of Things devices and wireless sensor networks is vital to understand (WSNs), because the Internet of Things has successfully combined WSNs, mobile network computing, and cloud server (web) technology—all of which have found practical applications in modern life. The purpose of this review is to discuss the various conversion methods that can be used as well as the available urban energy sources. In a smart city, this is how electricity is generated for linked IoT devices. As a result, the Internet of Things will be extensively used, self-sufficient, and have long-lasting energy sources to solve the issue of changing batteries. This is because it can be very challenging to replace exhausted batteries when IoT devices are utilized in hazardous or challenging-to-access environments [3]. We also talked about the significant savings on maintenance costs.

For the here and now and not too far in the future, with the arrival of 5G innovation in 2020 and the future 6G or past 5G, IoT and remote communications innovation generally, energy gathering solutions present numerous prospective advantages and exceptional components. Self-maintainable ability, omnipresent energy, a smaller carbon footprint, no battery replacement, and maybe no connection to power matrices are some of these benefits that cannot be offered by current batteries or framework-worked correspondences. They can also be carried with ease to hazardous, poisonous, and hard-to-reach environments. One of the applications for energy-harvesting techniques in our research is the Internet of Things (IoT), which includes, but is not limited to, the following: IoMT, which stands for Internet of Medical Things, IoMobT for Internet of Mobile Things, IoRT for Internet of Remote Things, and IoET for Internet of Environmental Things (IoEnvT). For the Internet of Things network to aggregate real-time data for automated smart cities, a consistent energy source from energy-collecting systems is required. The sources of ambient energy that can be produced in any city are examined in this review. Several researchers have described these sources in the literature. The benefits of energy collecting have frequently been discussed. In this analysis, our main objective is to demonstrate the local energy production that is possible for Internet of Things applications. As a result, based on the most recent studies conducted by a variety of researchers, we have examined and categorized a number of energy-harvesting systems that are currently available. All of the various energy transduction techniques have been considered, together with the output power that is available and, in certain cases, their efficiency. Our main classification criteria were the physical or chemical occurrences that surrounded the procedures used and their merit ratings. We were ultimately successful in producing our summary, which is displayed in Table 1, as the primary outcome of the review that was carried out. The harvester model type, the possible output power, and our information sources are shown in the table. Instead of having to replace the batteries in IoT nodes or gather energy from other sources to be stored in batteries, we came to the conclusion that energy could be harvested directly at the precise close vicinity to the application. Ambient energy is present almost everywhere there is vibration, heat, sunlight, wind, radio frequency, water, and a wide range of other naturally occurring sources [4]. This has led to the development of creative methods for making the Internet of Things technology usable and always available, as those shown in Figure 2. The IoT can receive the energy it requires from the EH to run constantly and everywhere.

**Table 1.** Types of sensors utilized in various smart city application divisions.

| Subcategory | Sensing Parameters | Type of Sensors | Distance GW-Sensor |
| --- | --- | --- | --- |
| Agriculture | Humidity, Temperature, Luminosity, Solar radiation, Soil, Conductivity, Ph | Ultrasonic Temperature Humidity Soil | 30 cm–15 km |
| Healthcare | Health signs | Biosensors | 3 m |
| Energy | Light intensity Motion Voltage Temperature Humidity | Temperature Humidity Motion | 15 km |
| Traffic | Motion Occupancy | Magnetic Ultrasonic | 500 m–1 km |
| Environment | $CO_2$, $NO_2$, $O_3$ Concentration Weather | Gas Temperature | 200 cm–5 km |

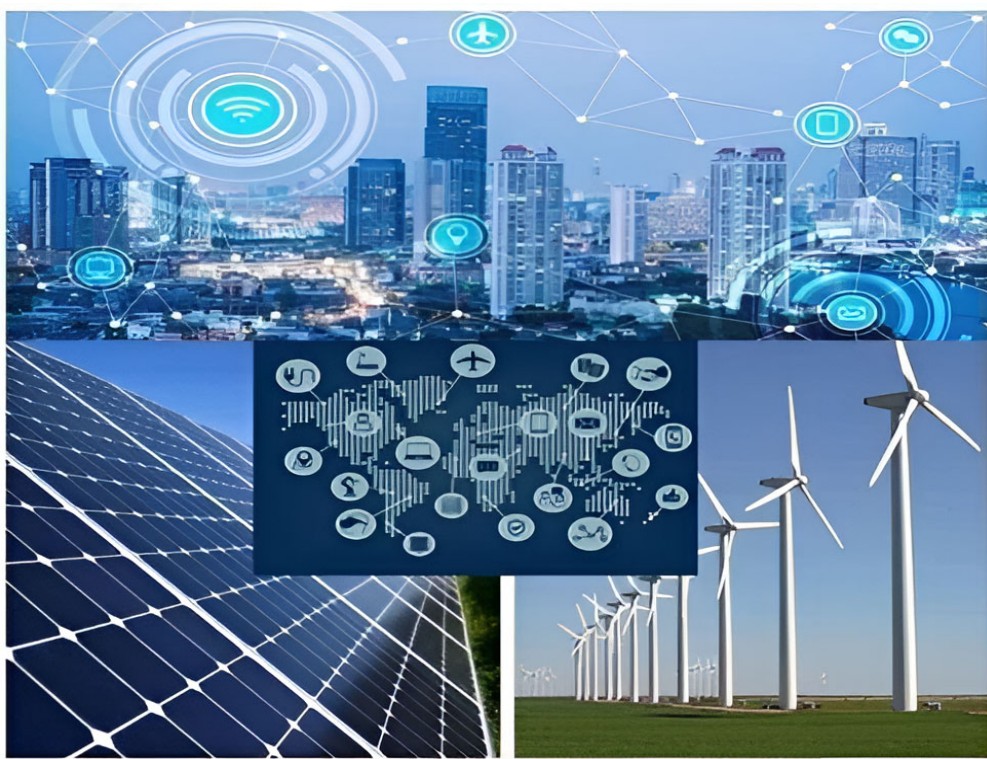

**Figure 2.** A smart city's energy-harvesting system.

The literature review serves as the cornerstone of scientific writing since it allows researcherd to familiarize themselves with the texts and identify the most notable authors who have written on the subject. For the literature review, we used a systematic analysis technique. "Energy harvesting", "Internet of Things", and "sustainable smart cities" were the search phrases used in the systematic review to look for publications in large databases.

This study investigates a variety of green energy-collecting techniques in relation to edge-based intelligent urban computing's smart applications for sustainable and smart cities. The five categories of energy-harvesting strategies currently in use are smart grids, smart environmental systems, smart transportation systems, and smart cities. The purpose of this paper is to provide a technical overview of energy-harvesting management strategies for green urban computing, including developed algorithms, evaluation standards, and evaluation environments. The authors' suggestions are:

1.  Discover a method to provide Internet of Things (IoT) nodes that have been placed with energy, which are both widespread and long-lasting.
2.  This study has looked at the majority of urban sources for energy that have been described by scholars in the literature.
3.  Energy gathering has been proposed as a solution, enabling IoT hubs to hunt for self-sustaining energy from ecologically wide sources.

The title of the final paragraph of this assessment is Contemporary Energy-Harvesting Techniques in a Smart City. Most methods scientists and engineers have used to convert energy from natural or artificial phenomena into an amount that can be utilized to power sensors and IoT nodes were described and categorized in this area. The majority of the realizable energies and, in certain circumstances, the effectiveness of the transduction system are listed in the section that follows in a condensed, simple-to-read table that summarizes our results. The research year and the information sources were mentioned. Our conclusions were based on research from roughly 10 years ago. We have demonstrated why, in the end, application-based energy harvesting is crucial for smart cities [5]. The requirement to collect energy from Internet of Things devices close to applications is clarified and supported in the last section.

## 2. State of the Art (SOA) of Energy Harvesting Schemes in a Smart City

The batteries of the IoT nodes were the main focus of earlier energy studies. By shrinking the size of the batteries and prolonging both their and the connected devices' lives, the entire network was created to be low maintenance. Reduced energy consumption of individual microprocessors, embedded sensors, and actuators utilized in IoT devices helped to achieve this in large part. Then, research was conducted to simultaneously reduce battery size, increase battery efficiency, and extend battery life. Numerous researchers believed that duty cycling and event-driven methods were crucial. They believed that by entering an idle state when not in use, some components might conserve energy. Despite all of these efforts, there will still be some maintenance costs associated with battery replacement and travel to the specific locations of IoT nodes. When IoT devices are positioned in dangerous smart city zones that are difficult to access, batteries will be difficult to replace. However, it will be more challenging if they are in hazardous regions and need rescue or emergency applications for smart cities. Every city has many ambient sources that can be exploited to scavenge energy, and there are many ways to achieve this [6]. The most common energy sources that can be acquired in a city are described in this overview, some of which include sources of thermal radiation, radio frequency, mechanical vibration, and solar radiation.

### 2.1. Mechanical Vibration Energy Harvesting (VEH)

Since human and/or mechanical activity is almost always present, mechanical ambient vibration energy can be effectively converted into electrical energy for IoT devices. Studies show that vibration energy can be generated by a variety of actions, including walking, a navy boot's heel striking the ground, an external transmitter, the vibration of a bridge, AC power lines, a bus or subway handrail, and more. The literature contains numerous reports of experimental and practical studies on vibration energy-harvesting (VEH) transduction. The four main subtypes of VEH mechanisms are turbine, electromagnetic, electrostatic, and piezoelectric [7,8]. In certain cases, combining two or more of these processes improved performance.

### 2.2. Electromagnetic Vibration Energy Harvesting

According to Faraday's Electromagnetic Induction Law, which claimed that an electromotive force (EMF) is produced when a conductor is inserted and moved inside a magnetic field, ambient vibration can produce energy in the form of electromagnetic transduction. This is completely shown in Figure 3. The literature describes a wide range of electromagnetic-based VEH energy-harvesting scenarios.

It has been established that wearable electronics may utilize the vibrational energy of a person's step. A 68 kg male may walk at a speed of two steps per second and a heel movement of five millimeters while producing 67 W of vibrational energy. However, a number of variables, like the gait of the EH device, electrical and mechanical power losses, and electromechanical efficiency, affect the effectiveness of these kinds of applications. This suggests that just a small portion of this energy might be effectively converted into useable energy. An experimental examination on the integration of electromagnetic and piezoelectric VEH was given by Song-Mao CHEN and Jun-hui HU. One hand-wound enamel copper coil and two pairs of NdFeB permanent magnets were the configurations for electromagnetic induction and the impact-induced vibration module for the piezoelectric harvester, respectively. They stated that the piezoelectric impact generated a vibration component that measured 429.3 J and the electromagnetic component that recorded 6547.2 J. The installation of a lead-zirconated titanate in the heel of a Navy work boot allows the collection of parasitic energy by heel striking, claim the authors Alireza Khaliah and colleagues in their review. They claimed that heel movement at a frequency of 900 MHz and a resistive load of 500 k could provide an average of up to 8.4 mW in a PZT bimorph structure. It has also been demonstrated that the vibration brought on by a person's breathing effort results in the apparent production of energy [9,10]. Shah Haidar, Ehsanul, and other researchers discovered that breathing can capture electromagnetic energy. They discovered that a resistive

load of 7 given by the armature resistance, when used to simulate the generator's current created by motions from respiration, produced a mean power of 2 mW and an estimated voltage of 200 mV, assuming a respiratory rate of 12 breaths per minute. This could result in the creation of wearable biosensors. Also, a five-minute workout generated 6.44 mW and 30.4 mJ of energy from an electromagnetic human respiratory EH, according to the study.

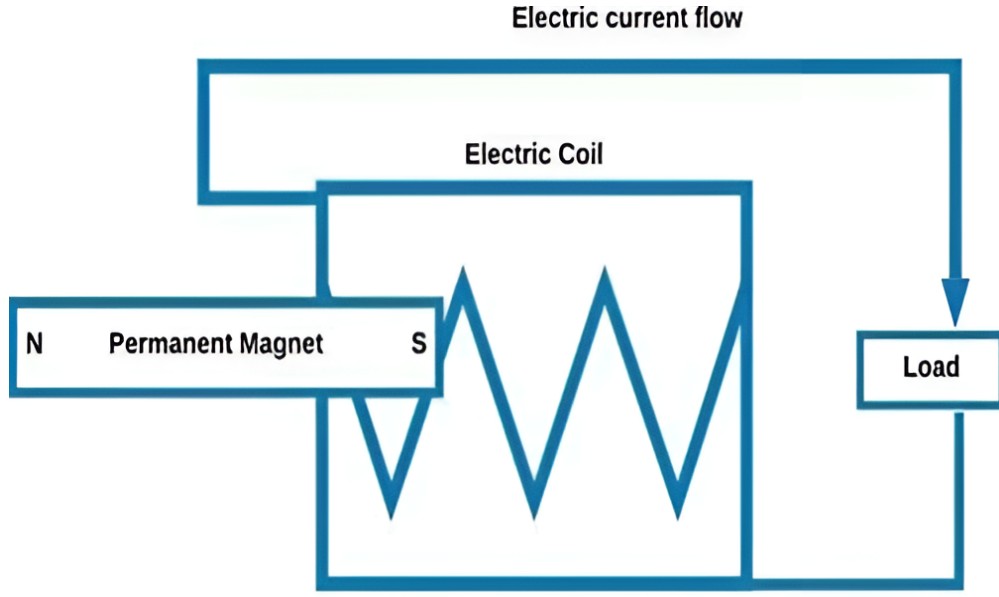

**Figure 3.** Electromagnetic Vibration Energy Harvesting.

## 3. Energy Harvesting Management for Intelligent Urban Computing

In order to achieve sophisticated and extensive intelligent urban computing while reducing energy and power consumption in the IoT ecosystem, energy-harvesting strategies are combined with the best on-time efforts and ideal solutions [11]. A technological taxonomy is used to organize energy-harvesting methods on platforms for intelligent urban computing in this part. Our proposed taxonomy for energy-harvesting systems, derived from the literature review, is shown in Figure 3. This area discusses energy-harvesting methods, such as supervised and unsupervised machine learning algorithms, fuzzy logic and methods, deep learning, and evolutionary and naturally inspired algorithms. On the other hand, the intelligent urban computing platforms covered in this part, can be categorized into four main categories: smart grids, smart homes, smart cities, and smart environmental systems [12–14].

### 3.1. Smart Home

One of the most common issues with energy harvesting in IoT installations is the smart house. The two sections are management operations and assistance operations. A smart home's assistive operations sight provides users with basic assistance with daily tasks. Also, management tasks provide special capabilities for smart houses [15]. Monitoring the home's energy efficiency and controlling the lights and appliances to consume less energy while still meeting the demands of the occupants are a few examples of these activities [16]. As seen in Figure 4, some components of a smart home can be set up automatically: Each light bulb in your home can be controlled by a simple remote control. The light's brightness and intensity may be adjusted, and it can be programmed to turn on or off automatically after a predetermined period of time. All of the switches in a smart home can be remotely turned off. With timers, you may plan when specific devices turn on and off. One of the most crucial and critical components of a smart home is a set of home security cameras.

Each activity occurring within the house can be observed with the help of sophisticated sensors. You can control who has access to your home and how your doors and entrances work digitally with the aid of intelligent entrance software. A smart home's automated thermostat is capable of independently adjusting the temperature. Finally, smart lighting and door lock systems can be employed in smart homes to reduce energy consumption and improve energy efficiency. This category includes four studies that categorize current energy-harvesting methods for IoT applications pertaining to smart homes. Participants in this study [17] described a sophisticated system for monitoring patient and home health. An IoT system has been used to carry out this study. Sensor nodes can wirelessly transfer low-power data to system gates using information such as body temperature, heart rate, galvanic skin response, environmental data, and contextual data. Radio and solar energy-generating techniques have been used to power the sensor nodes. Also, the generation of energy using rectifier circuits, radio frequency, and multiband antennas is illustrated. The examination of a room sensor that draws power from a photovoltaic (PV) solar cell reveals that both the generated voltage and the amount of existing voltage were sufficient for installing the sensor nodes.

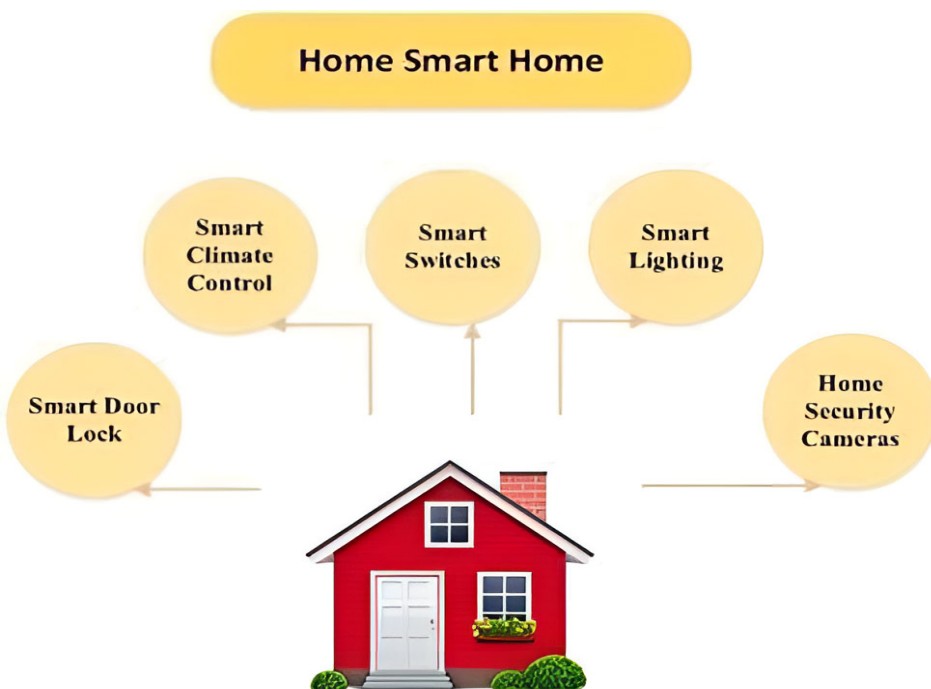

**Figure 4.** Smart Home.

For embedded, battery-free sensors focusing on smart houses, The use of thermoelectric strength gathering was advised. Similar implementations include vacuum-isolated plates for structures with battery-free tension feeling. The plates could be used to force a wireless detector to determine their strain grade because they significantly increase heating and cooling power consumption. Beginning with validated climate data, they constructed a prototype for the available strength. They then combined the thermoelectric generator with a vacuum-isolated plate and placed it in a window for examination in order to gauge its substantial electrical strength. The authors discovered that the projected temperature slope might eventually be used to run a sensor node. They demonstrated how thermoelectric power collection could enable a special class of implanted, battery-free, low-maintenance sensors for smart houses. The ground tiles developed by [18] can be used anywhere in the house and can produce enough power when a person stands on them to transmit data wirelessly to an electronic device. The best output zone for power extraction is along the octahedral–tetragonal step border, which has an immediate advantage for balanced power

generation. The authors also developed a portable robotic piezoelectric transducer using a foot switch. A wireless transmission scanner network and the main appliance's transmitter change device collaborate to effectively regulate the feedback impedance that a person encounters from their steps. According to the research, the readily adaptable tiles offer hope for a smart home for people in the future, with potential applications for those with mental illnesses. Ref. [19] Have developed a trustworthy smart house switch mechanism that combines wireless connectivity for smart sockets and ports, access to the home electrical network, and power generation and processing for advanced electric modules and circuits. The concept and development of communication networks, as well as ego energy storage safe fluorescent signals for smart homes, are covered in this article. The electrical energy used by a building is made extremely safe thanks to the device architecture presented in this study. Even better, the plan collects and stores energy so that any digital devices in use can use energy buffer solar panels. In order to use less base energy, the authors suggested a smart house that leverages wireless ZigBee connectivity for energy management services. For controls, automation, and ease of use, the built-safe smart house used Message Queuing Telemetry Transport (MQTT). The structure's safety and energy efficiency have improved, according to the findings of extensive concept tests. According to Figure 5, having smart cities is one of the most interesting and challenging circumstances for a smart environment. They include lighting, energy, public health, building, housing, and education. They also encompass a wide number of sectors. To illustrate how various approaches to energy harvesting for Internet of Things (IoT) applications might be grouped, eleven research studies are provided in this paragraph. In [20] suggested a design that can handle applications with large amounts of data and are up to the task. They have combined the IoT idea with cloud computing and mobile computing strategies to address this issue. The ability to manage enormous volumes of data from images and videos was the key objective of this design. The benefit is that all three layers can access the information they need to conduct their tasks separately without having to communicate with one another. Making this notion n layers deep and including different applications is one of the author's objectives [13].

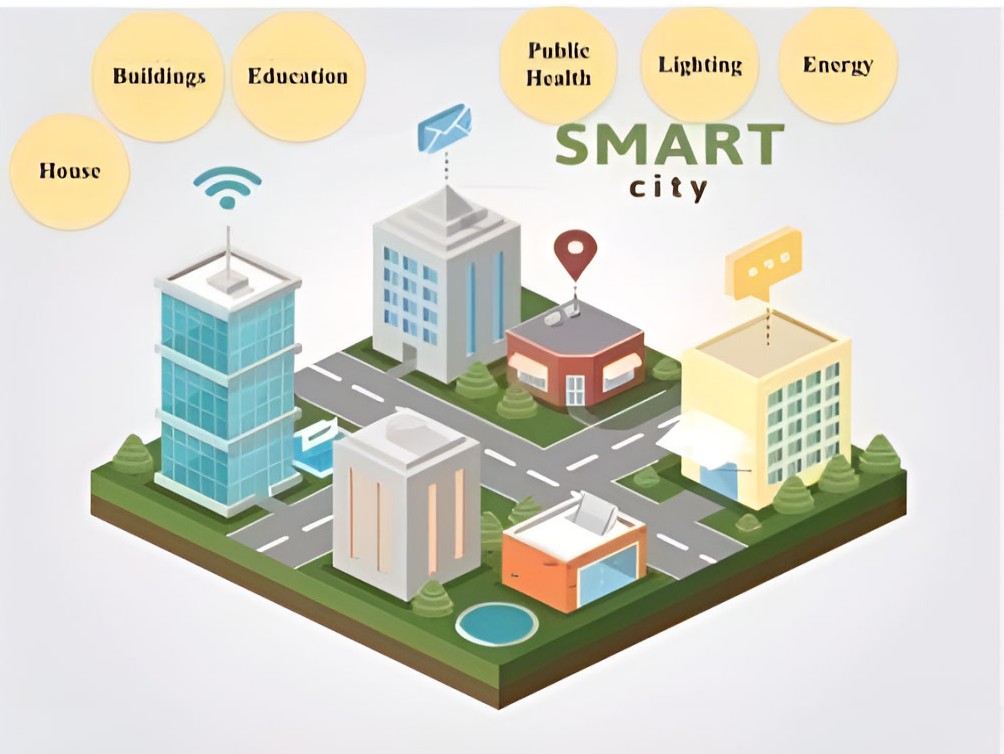

**Figure 5.** Smart City.

A resource allocation technique that equally splits the available resources between radio and communication has been described by [21]. The time average rate for maximizing computation used by the authors improved the system's performance. The recommended method has enhanced dynamic power decision-making, edge work loading, and bandwidth. As a result of WET, energy-harvesting devices may now temporarily store RF signals on their own bodies as batteries and then transfer that energy to other devices when their own batteries are depleted. The fundamental concept is to use the Wireless Energy Transfer technique to gather signals like radio frequencies as energy and release them in needle states, allowing UEs to utilize their optimal system computing capacity and rate without taking into account the energy sources that are necessary for UE. For massive fog computing networks, a ground-breaking dynamic network slicing design was put forth by [22]. This article is relevant to fog computing systems, which allow cloud data centers to expand by a substantial number of fog nodes scattered over a considerable area. Each hub must obtain energy from the local environment in order to offer computational services to the neighborhood. The authors describe the concept of dynamic network slicing in this section. Theoretically, a geographical orchestrator distributes workloads among nearby fog nodes by precisely allocating energy and computing resources in order to offer a specific type of service with accurate quality of service assurances. Each section's resources can be dynamically modified to meet service requirements and energy availability. Using actual BS location data from a system with more than 200 BSs in Dublin, the functionality of the authors' notion is developed and its substructure is evaluated. Scalar results show that the authors' substructure significantly enhances the workload following qualification of fog computing.

For autonomous ICT applications, Modern scavenging techniques were detailed. When they looked at better power sources and alternatives like paper, RF signals, and energy gathering technology, the shortcomings of conventional sensor node power sources like batteries were brought to light. In order to track the sensor nodes in real-time and capture their orientedless energy-scavenging activity, the researchers developed a three-dimensional RFID cubic antenna made using inkjet technology. A wireless sensor transmitter prototype was then created to demonstrate the inerrability of the scavengers, and Satao's Stargate 64 Antenna Chamber system was used to measure the module's radiation diagram. They evaluated the efficiency of their devices and calculated their frequencies for energy collection using a variety of locations, the bulk of which were in Tokyo, including train stations, laboratories, city streets, and open areas. The results of the experiment show that the radiation emission of broadcasting devices is frequency dependent. Charts also show how much energy may be obtained from a Tokyo worker's usual day. An Omnidirectional Biomechanical Energy Harvesting Block (OBEH) that may generate power from human movement was proposed by [23]. Simply by moving about, humans use a significant amount of mechanical electricity every day. The age of the gleaming city will soon be available, thanks to IoT growth. After all, chemical batteries, which are used in wireless sensors and similar applications, have several disadvantages. WSNs must operate more sustainably in order for the smart city aim to be achieved, and a self-generating power grid should be installed as a protective precaution to reduce reliance on batteries. An effective design strategy for the OBEH walkway block is described in this article. It consists of three main parts: tracking footprints is dealt with in the first, modeling output results is dealt with in the second, and dependability and optimization of the OBEH walkway block are dealt with in the third. The first variety requires a long time and the second size and bearing of strides are two unique forms of haphazardness that are detected by this inquiry. Using the Dependability Mindful Planning problem, the outcome energy given by an annular piezoelectric layer attached to the primary layer's core was increased while reaching the firm goal quality of 99.87%.

A 5G support organization approach for cognitive IoT networks with RF power gathering was presented. The goal of this framework is to reduce the amount of control energy used by each IoT sign while also increasing production and convincing clients of the value of the services provided. The outcomes show how similar the simplex technique in the

suggested structure is to voracious computing. This group will focus on the NOMA-based stock management zone for its next IoT projects. The idea of combining a Wi-Fi access system with the capabilities of the Wireless Sensor Network (WSN) was put up in order to create a unified public network system that would, in large part, encompass components of any of these issues (2012). By extending organizational chains from information identification at specific IoT device levels to the distribution of end-user products and perceived information security via the group's Wi-Fi access network, the proposed device's interdisciplinary design enables the end-to-end approach. Due to the several steps required to keep the energy consumption of the WSN protector's components under control, the suggested construction can be considered ecologically sensitive.

With a virtualization sensor and a green, decentralized justification design intended to reduce overall network energy consumption, the WSN system is heavily concerned with environmental issues. Users may access the internet from anywhere thanks to the city-scale Wi-Fi connection point content aggregation, which also includes user-side data use and automatic content distribution. The collection of real-time measurements pertaining to the atmospheric characteristics of actual artifacts is another feature of the WSN concept.

*3.2. Bright Grid*

The software develops smart grid gadgets with various features and remote adaptability. The conventional inactive, centralized model of energy production and sharing can now be replaced in the smart grid with a dynamic, bidirectional, decentralized one that is implemented closer to the user and at the edge. As we go closer to the smart grid, we must realize that it is software and its form, not the physical principles themselves that need to be handled. The combination of renewable energy and energy optimization is essential for both smart grid mitigation and sustainable energy harvesting, as seen from Figure 6. For smart metering, smart charging, piezoelectric energy harvesting, and power-electric flow management, the energy-aware smart grid can use IoT technology. Smart grids can employ IoT to improve energy efficiency, increase the proportion of renewable energy, and minimize the environmental impact of energy use. This category includes six studies that categorize the current approaches to energy harvesting for IoT applications' smart grids. This service has been offered by applying piezoelectric energy harvest-ing to build a smart city. Vibrational energy is converted into electrical energy by piezoe-lectric materials. Titanium lead zirconate (PZT), a nonconductive material, is a compo-nent of piezoelectric devices.

It is attached to a base and rests between two metal plates. While being more expensive than other forms of power generating, piezoelectric power generation has no adverse effects on the environment. In this study, switch bias is removed by using a piezoelectric vibration oscillator to generate a sine wave at the desired frequency and voltage. This renewable, lossless system has higher energy efficiency. Examples of the system's low consumption uses include agriculture, body networks for medicinal purposes, and wireless sensor networks. Ref. [24] Provided a framework for a plan that aims to reduce energy use while protecting client privacy. The goal is to calculate how much energy each person needs and how much energy was lost owing to interference from nonsuppliers. They were able to evaluate the energy effectivity and client security since they introduced a rank for each of the features, such as energy dissipation and data security [25]. They also learned that utilizing a device to obtain information boosts energy use and also obtains the client's secret while saving more energy. This research had the advantage of looking at rechargeable batteries with limited storage capacities and discovering a device called the EH that could produce discrete amounts of energy at each instant in independent and identically distributed distribution. By storing excess energy for later use, these batteries effectively prepare energy. They can also enhance privacy by concealing the use of each electrical device from the manufacturer. Ref. [26] investigated a variety of concepts and tools for utilizing body energy to power assistive technology. Their study motivated them to create a ground-breaking method that outperformed earlier technologies by increasing power output and extending the battery life by utilizing a variety of transducers.

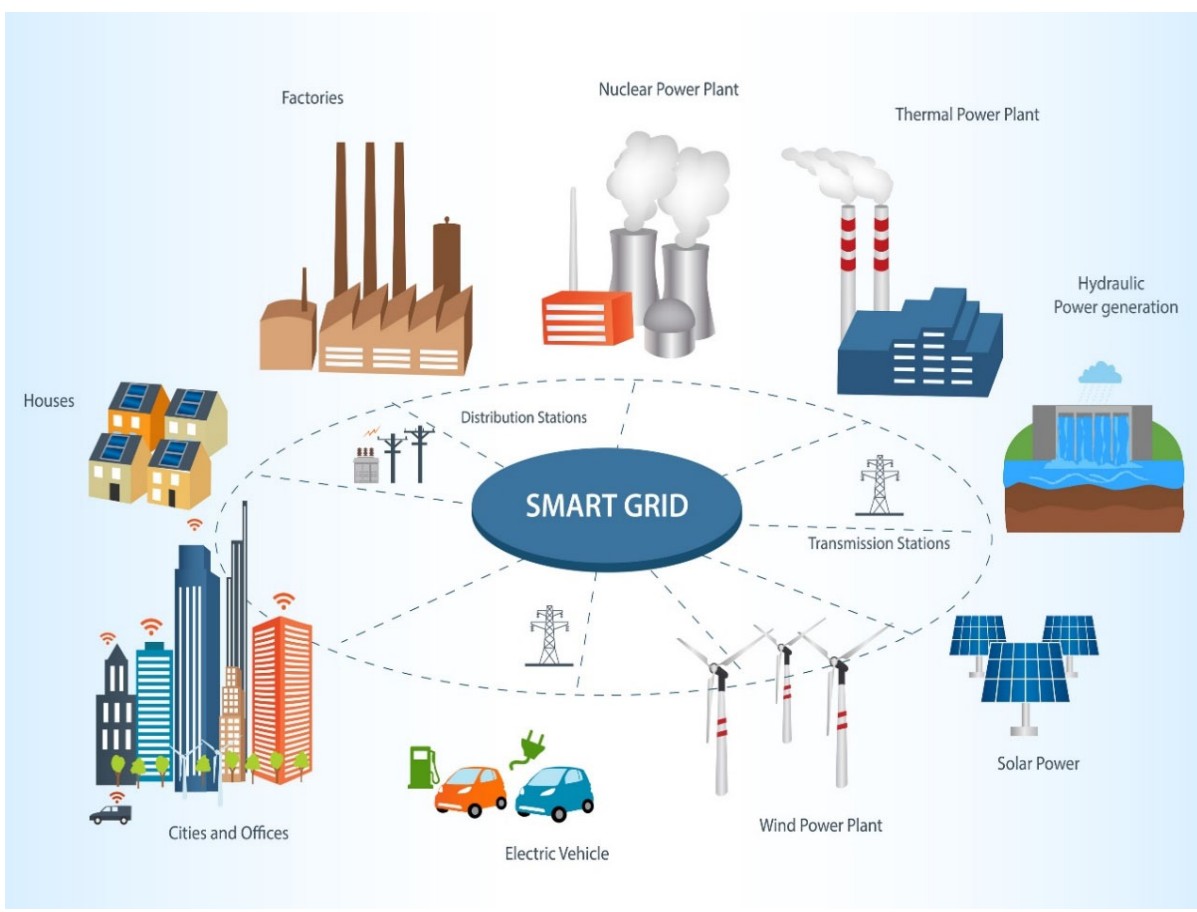

**Figure 6.** IoT smart grid applications for intelligent urban computing.

The authors assessed their work using piezoelectric generators mounted on people and DC-DC converters. Although if it is currently impractical, the devised plan has the potential to be useful because it suggests a novel strategy. An IoT-enabled plan for enhancing the smart grid's usefulness on the internet of things was presented by [27]. In this framework, the nodes are cordless IoT widgets and internet-connected system-generating elements. By utilizing incompatible channels and cognitive radio, this construction overcame a spectrum shortage, improved band performance, and calmed the rowdy states of a band. The energy-scavenging technique, also known as EH, improved a number of factors, including the longevity of the cordless widget and the issue of limited supply. The authors, as a result, published an advanced networking architecture for the Internet of Things (IoT) named an energy-harvesting IoT-enabled cognitive system. Examining energy consumption stability and EH features to permit battery-free operation will be the main objective of the upcoming development. The technique does not restrict the devices' use, movement, or the number of charged devices; rather, it prohibits the devices from being recharged by attaching a dongle to them. To identify devices that have this capability and are within the authorized range, this system uses communication between devices. The battery's remaining capacity is taken into account while setting priorities. The energy transfer operation is performed using the resonant coupling circuit. The findings of the simulation indicate that it would likely take between 10 and 20 s to connect to the charging system. Also, using high-quality amplifiers improves performance significantly and does so regardless of the user's mobility. Benefits of the system include the ability to be used both indoors and outside, the freedom to move about while charging, prioritizing charging based on how much juice is left in the device, and offering a premium version that enables users to prioritize charging even more. Future applications of the proposed technology could include electric cars and other mobile gadgets. A radiofrequency-based energy distribution and power storage system has

been proposed. The recipient device must have a sizable antenna that can gather energy for this method to work. Here, reducing energy use and improving energy efficiency are the objectives. Programmatically speaking, merging the subcarrier and the power consumption component is difficult to achieve this purpose. The results of the experiments show that this innovative power depletion mechanism speeds up energy supply and reserve, especially when there are more users than devices communicating with one another.

## 4. Smart Environmental Systems

The actual environmental problems we are currently experiencing are largely a result of poor air quality and contaminated water supplies. A healthy community must be maintained on the planet in order to allow for sustainable growth. Thanks to the advancement of better sensors and Internet of Things (IoT) technologies, environment monitoring has evolved into a smart environment monitoring system in recent years. The "smart environment", a cutting-edge technology, offers a variety of instruments and answers for a variety of environmental issues, including resource waste management, air and water pollution, and other environmental indicators. The actual environmental problems we are currently experiencing are largely a result of poor air quality and contaminated water supplies. A healthy community must be maintained on the planet in order to allow for sustainable growth. Thanks to the advancement of better sensors and Internet of Things (IoT) technologies, environment monitoring has evolved into a smart environment monitoring system in recent years [28,29]. A study on the inline vertical cross-flow generator for future hydroelectric power harvesting within water inventory channels to power water measurement instruments was presented. The water generation device's block structure forms were computationally examined to discover the optimal structure. The observational findings demonstrated the validity of the computational method employed in this study to assess the effectiveness of this micro-water producer. For the administration of smart building microgrids, a reliable control strategy for solar energy harvesting was developed by [30]. The suggested method makes use of parallel and distributed computing, as well as augmentation and control computations. A microgrid with a power velocity tracking system and power velocity array renewable energy sources is installed on the roof of a 12-story building and serves as the testbed for the hardware implementation of the proposed technology. The suggested design has been demonstrated to be effective in experiments, and the necessary robustness and reliability have been reached. Due to their capabilities for quick self-recovery and omnipresent computing, these built, IoT-enabled, dependable control systems are energy sustainable and completely tolerant. Computational Liquid Elements (CFD) simulations and AI computations like Fake Brain Organizations (ANN) have been employed to reproduce the complex 3D stream field of the turbines. Simulink, a MATLAB program, is used to build artificial neural networks, and ANSYS is utilized for simulation. The findings indicate that using RWTs for energy will increase both animal and human safety, reduce noise, and use more recyclable and durable materials than conventional turbines. The turbine performs better than other turbines in terms of Power Coefficient as well (Cp). An effective method for harnessing solar panels to extend the life of a fog computing network has been presented. Since batteries are frequently used to power fog nodes, finding a technique to manage their energy effectively can help them survive longer.

They used an Energy-Effective Calculation Offloading (EECO) methodology to categorize the hubs according to cost and energy consumption before assigning them their offloading requirements. By outsourcing computing to edge servers, smart energy management can predict how much energy the nodes will consume. In comparison to fog networks powered by solar panels, the researchers' technique increased network lifetime by 20% and 100%, respectively. Ref. [31] Solution to the finite vigoro layout problem involves using solar energy to replenish the WSN's battery. The ZIGBEE network, which is used in this technique, enables energy collection even in older WSN iterations. The Net Sim simulator was used to run the simulations. The simulation's results demonstrate that the suggested technique mounts every parameter utilizing a drawing technique that is

appropriate for the direction and placing substrates themselves (application layers two and three), optimization of the algorithm to reduce the energy required for the proposed method, and releasing a more practical IoT- and vigoro-based platform for flexible farming monitoring (Figure 7).

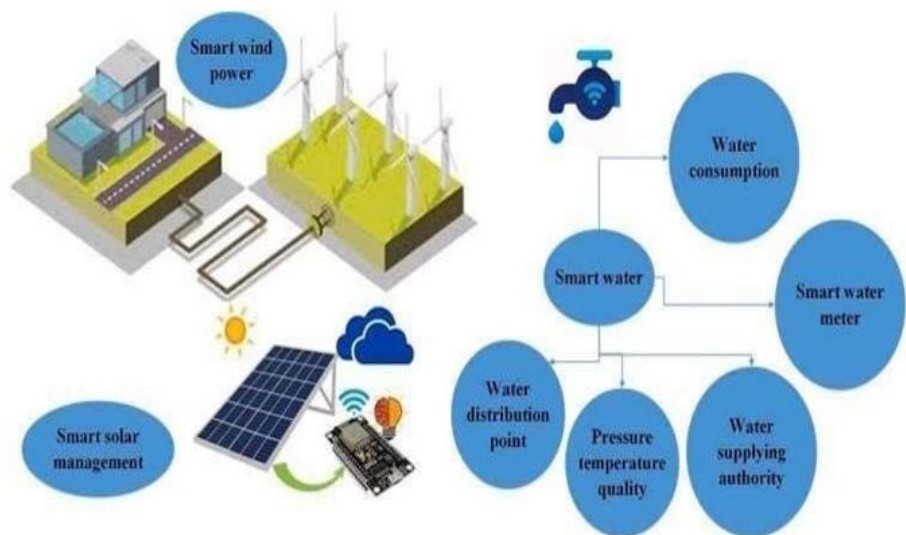

**Figure 7.** Smart environmental systems intelligent urban computing in IoT application.

*4.1. Issues around "Smart Energy" for an Effective and Efficient Energy Supply Are Currently Being Debated Extensively in the Most Recent Publications*

4.1.1. Systems That Use Smart Energy and Smart Energy

The names "Smart Energy" and "Smart Energy Systems" have been popular in recent years to describe a strategy that goes beyond what is meant by "smart grid". Smart Energy Systems take a more integrated, holistic approach to the inclusion of more sectors (electricity, heating, cooling, industry, buildings, and transportation), as opposed to smart grids, which primarily focus on the electricity sector. This allows for the identification of more feasible and affordable solutions to the transformation into future renewable and sustainable energy solutions. This essay begins with a survey of the relevant scientific literature. The term "Smart Energy Systems" is then discussed in relation to the concerns of definition, solution identification, modeling, and storage integration. The concept of a smart energy system, it is concluded, represents a scientific paradigm shift away from single-sector thinking and toward an understanding of coherent energy systems that demonstrates how to profit from the fusion of all infrastructures and sectors [32].

4.1.2. A Review of Polish Urban Development Plans Examining Smart Energy for Smart Cities

Understanding the current state and future advancements of smart energy techniques is crucial for the effective and efficient supply of energy to meet the needs of the exponentially increasing energy demands of modern cities. Smart energy is a vital component of the notion of a "Smart City". Using content analysis methods, this research examines how the Smart Energy agenda is included in Polish plans for developing smart cities. The most often mentioned elements of the Smart Energy agenda were the stakeholders' involvement, spatial aspects, Smart Energy conceptions, and Smart Energy key sectors. Universities, small enterprises, and public governance organizations are all essential key actors covered by stakeholders' involvement in Smart Energy agendas. The individual, city, regional (subregional), national, and international (EU) levels are included in the spatial dimension components of the Smart Energy agenda, with the city level naturally dominating. The component for "Smart Energy conceptions" demonstrates a stark difference between "peripheral" Smart Energy conceptions and the four "core" Smart Energy conceptions

(renewable energy, energy efficiency, energy-saving technology, and energy security). It was discovered that the only sectors mentioned in the analyzed urban development plans with relevance to the Smart Energy agenda were the building, transportation, lighting, and manufacturing sectors. The study's findings aid in a better knowledge of the Polish smart city and energy planning environments and may serve to enhance the cities' spatial planning tactics [33].

### 4.1.3. Taking into Account Energy Costs and Carbon Emissions, Stochastic Operation Optimization of the Smart Savona Campus as an Integrated Local Energy Community

Sector coupling increases the penetration of renewables and lowers carbon emissions while aiming to integrate various energy sectors and use of synergies resulting from the interaction of various energy carriers. Sector coupling fits in well with the idea of an integrated local energy community (ILEC) at the local level. In an ILEC, active consumers make decisions about how to best meet their energy needs by coordinating the use of a variety of multicarrier energy technologies. This results in greater economic and environmental advantages over the status quo. This article examines the stochastic operation optimization of the University of Genoa's smart Savona Campus in light of financial and environmental considerations. With two electrically connected multienergy hubs that use absorption chillers, electric and geothermal heat pumps, solar thermal, combined heat and power systems, PV, and solar thermal, the campus is treated as an ILEC. With the right bidding tactics, the ILEC can take part in the day-ahead market (DAM) from this perspective. The fast forward selection algorithm is used to preserve the most representative scenarios while lessening the computational load of the following optimization phase. The roulette wheel method is used to generate an initial set of scenarios for solar irradiance. In order to optimize the operation strategies of the various technologies in the ILEC as well as the bidding strategies of the ILECs in the DAM, both energy costs and carbon emissions are taken into account through a multiobjective approach. This is accomplished through the use of mixed-integer linear programming (MILP). Results from a case study demonstrate how the ILEC's ideal bidding techniques on the DAM enable minimization of the users' net daily cost. Similarly to environmental optimization, the ILEC also runs in self-consumption mode [34].

### *4.2. Implemented IoT Technologies in Smart Cities*
### 4.2.1. Different Types of Sensors Used for Smart Cities

By 2050, 85% of the world's population, according to experts, would reside in urban areas. Cities should therefore be ready to meet the needs of its residents and offer the greatest services. The smart city, a more effective system that maximizes its resources and services via the use of monitoring and communication technologies, is a typical representation of the idea of a future metropolis. Making the shift to smart cities is thus one of the measures that cities all around the world can take to become more sustainable. Here, sensors are crucial to the system because they collect pertinent data from the city, its residents, and the accompanying communication networks that transmit it in real time. Although there are many applications for these sensors, they can be divided into six categories: energy, health, mobility, security, water, and waste management. This review includes an overview of several sensors that are frequently utilized in initiatives to create smart cities based on these groupings. There are insights regarding various applications and communication technologies as well as the primary potentials and difficulties encountered when converting to a smart city. In the end, this process is about more than just smart urban infrastructure; it is also about how these new digitalization and sensing advancements enhance quality of life. Smarter communities are those that invest in, socialize with, and adapt to these technologies in accordance with local and regional societal requirements and ideals. Privacy and disruptions to cyber security continue to be major vulnerabilities.

### 4.2.2. Table That Lists Several IoT Networks Used in Smart Cities, Such as LoRa, WsNs, Bluetooth, etc., and Displays Their Benefits and Drawbacks, as Well as Energy Usage, etc.

IoT networks used in smart cities, such as LoRa, Bluetooth, Wireless Sensor Networks (WSNs), and others, are listed in the following table (Table 2), along with details on their advantages, disadvantages, and energy consumption:

**Table 2.** Several IoT networks used in smart cities, such as LoRa, WsNs, Bluetooth, etc., with their benefits and drawbacks, as well as energy usage.

| IoT Network | Advantages | Disadvantages | Energy Usage |
|---|---|---|---|
| LoRa | - Long-range communication<br>- Low power consumption<br>- Scalability | - Low data rate<br>- Limited bandwidth<br>- Not suitable for real-time applications | - Low power |
| WSNs | - Scalability<br>- Low power consumption<br>- Suitable for sensor data collection | - Limited range<br>- Network topology maintenance<br>- Limited processing capabilities | - Variable, depends on node activity |
| Bluetooth | - Low power consumption<br>- Wide device compatibility<br>- Low cost | - Short range<br>- Interference in crowded environments<br>- Limited scalability | - Low power |
| Zigbee | - Low power consumption<br>- Scalability<br>- Suitable for home automation | - Limited range<br>- Interference from other wireless devices<br>- Limited data rate | - Low power |
| NB-IoT | - Wide-area coverage<br>- Low power consumption<br>- Suitable for large-scale deployments | - Costly infrastructure deployment<br>- Limited bandwidth<br>- Relatively higher device cost | - Low power |
| 5G | - High data rates<br>- Low latency<br>- Supports massive IoT devices | - Infrastructure deployment cost<br>- Limited range<br>- Energy-intensive for small devices | - Variable, depends on usage |

Note that this table gives a broad overview of the advantages, disadvantages, and energy consumption of several IoT networks. Depending on the implementation, hardware selection, and deployment conditions, the actual performance and energy use may differ. Planning IoT deployments in smart cities requires careful analysis based on the particular needs of your project in order to select the best network technology.

## 5. Smart Transportation Systems

One of the most important aspects of the design of a Smart City is efficient transportation. A strategy for innovation effectively handles both new and existing transportation infrastructure, putting an emphasis on functional viability, securing insurance, and minimizing the need for individual cars at a low cost. Additionally, it provides a ground-breaking plan for the adoption of numerous forms of transportation, better support, and expert traffic control solutions [35]. Figure 8 illustrates how IoT in transportation makes travel safer, greener, and more comfortable in addition to assisting travelers in moving from one place to a better one. A smart car, for instance, integrates communication, entertainment, navigation, and safe, efficient transportation. Thanks to the Internet of Things, travelers may easily stay connected to all forms of transportation, where can the vehicle connects to the internet through Bluetooth, Wi-Fi, 3G, 4G, other vehicles, smart traffic systems, and a variety of additional wireless connections. In order to avoid collisions, a vehicle's interior or exterior sensors can issue lane departure warnings and continuously scan for objects on both sides. With the use of the Internet of Things (IoT), which goes beyond cars, it is now possible to automate real-time decision-making to improve travel. In order to categorize

energy-harvesting management solutions for IoT settings' smart transportation systems, we looked at eight studies in this part. A compressible ball-twine electromagnetic power harvester (MMR)-based mechanical movement corrector has been designed, modelled, tested in-lab, and its findings have been described [28,29,36–38].

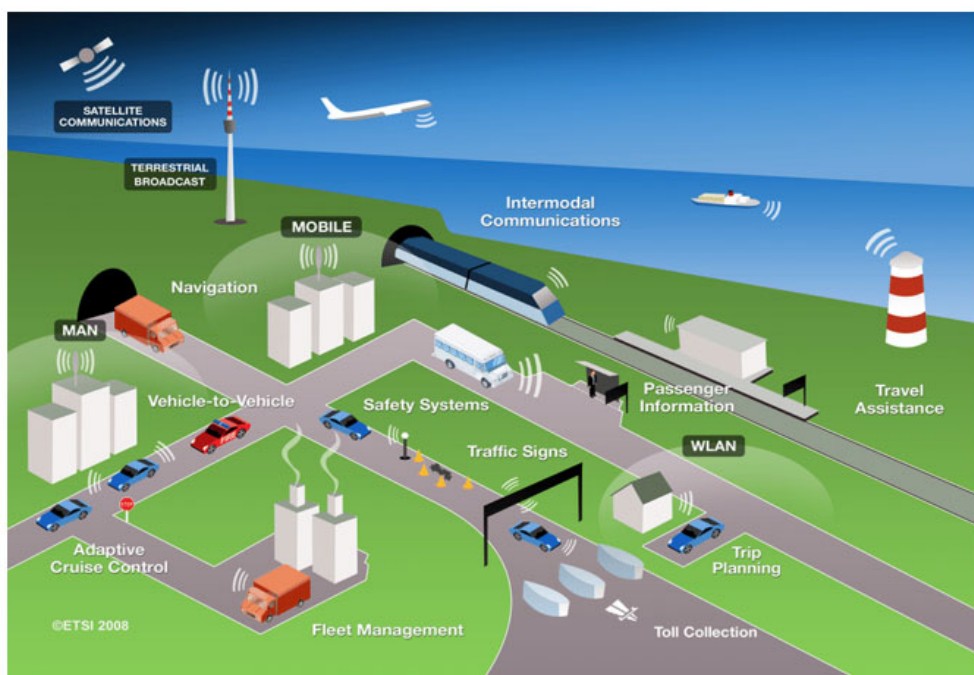

**Figure 8.** Smart transportation for intelligent urban computing in IoT applications.

To obtain skilled railroad forwarding and in order to dissect the kinetic specification of the connected device and foresee the power harvesting implementation of the harvesters at various train rates, a universal pattern thinking of the train-rail-harvester interaction was expanded. In-lab and field experiments were conducted to gauge the efficiency of the power harvesters and implement the strategy. The results of the ground experiment indicated the presence of mean energies of 2.24 W and 1.12 W. In addition to precomputed buffers and a tunable moderator keyed by a foreign resistive bar of the manufacturer, it is demonstrated that the suggested ball twist-based power harvester functions as a stable more inert power harvester when the one-way clutch is activated. Additionally, a novel kinetic energy harvester-based mode-detecting device for transportation was exhibited. (KEH). The attention-based Long Short-Term Memory serves as the foundation for this detection method (LSTM). The classification framework is compared to the machine learning techniques K-Nearest Neighbor (KNN), Naive Bayes (NB), and Support Vector Machine using a 38.6 h dataset (SVM). Comparing the recommended KEH-based technology to more advanced transportation mode recognition systems, the analysis shows that it uses less power and has a detection accuracy of over 97%. The size of the proposed prototype is a disadvantage because it is similar to that of common mobile devices. Mob Eyes was useful software created specifically to limit citizen surveillance, provoke challenging movement to spread information across local media, and produce a chip list to cast doubt on monitoring data. It makes use of analytical patterns to judge Mob Eyes' formality, its usefulness as a means of transportation, the outcomes of the simultaneous application of several harvesting aspects, the value of net aerobic, and its dependability in an alluring civil exploration use. The MATLAB program was used to evaluate the model. They reached the conclusion after looking at the data that Mob Eyes might be shaped to achieve the perfect balance of total and aerial privacy. A ground-breaking method for energy harvesting has been published. The authors were able to achieve his goal of making advertising structures on haulage routes dependent on power networks by using piezoelectric construction

materials [12,25]. They admitted that they had only looked at a portion of the proposal and that more investigation would be required before they could determine its potential. They suggested using the high wind to generate the required power for billboards amid a traffic jam for upcoming projects. Ref. [39] Have displayed a prototype of an energy strategy that uses electromagnetic vibrations to power railside gadgets. The authors considered three elements in order to make sense of their undertaking. One of the best aspects of their idea is their system, which consists of sensors and an energy collector that function dependably in all conditions. Hence, both tunnel railway and urban rail transit can be accomplished using this technique. The authors tested their methods using two-wheeled models. The first model's wheels are spherical, whereas the second model's wheels are not. The author's methods are suitable for assessing rail vibrations when using OOR wheels, according to experimental tests. The offered methods are both economical and environmentally friendly. Boyle and colleagues studied battery longevity and durability, human robustness's susceptibility to radio frequencies, connection mechanisms, and sender fountainheads in a variety of platforms by tracking batteries using a data mule agent and Wireless Sensor Networks (WSN). The authors suggested a novel strategy to improve the aforementioned characteristics. Yet since there was only one retention appliance, it was only a hypothetical situation. Ref. [40] demonstrated how they can use "piezoelectric and permanent-magnet" to create energy from the speed of cars on speed-bumps in order to apply energy-harvesting technology. In this paper, they used a number of meta-heuristic strategies. To provide this energy, a recurrence motor was combined with proper programming for evaluation measurements [41,42]. The metaheuristic algorithms of the MATLAB framework should benefit from it. According to the study's results, this technique not only maximizes energy but also reduces 9 tons of greenhouse gas emissions. The results of all experiments show that the generator only converts 85% of mechanical energy into electrical energy, or a total of 700 W.

## 6. Discussion

We use technical reports to respond to these enquiries in accordance with the prepared questions in Section 3 as follows: What intelligent urban computing models for the Internet of Things are being investigated in terms of management tactics for energy harvesting? Figure 9 illustrates how much energy harvesting has already been carried out for Internet of Things models of intelligent urban computing using eight futuristic transportation options. There are four studies each on smart home and environmental systems, as well as six studies on the smart grid. Smart houses and smart environmental systems each have four research articles, suggesting that both techniques still face two unresolved problems even though smart cities have a higher impact on the management of energy harvesting for intelligent urban computing in IoT. To lower the energy and power consumption of IoT devices, established metropolitan areas are studying and implementing smart solutions for smart city environments. RQ2: What optimization strategies and models are investigated in this literature? In light of Figure 10, the heuristic-based technique is used more frequently than the alternatives. The developers used heuristic calculations to evaluate how effectively smart homes, smart cities, and smart transportation systems gathered energy in accordance with the specified scientific classification. Studies on smart transportation have examined energy management using fuzzy methodologies based on fuzzy logic models. The authors evaluated the current energy-aware solutions in case studies, including smart cities, using fuzzy, heuristic, and natural-inspired algorithms. For the management of smart grids, authors have used supervised machine learning methods, heuristic algorithms, and strategies derived from nature. Then, a list of useful optimization techniques is created for efficient environmental management. Fluffy tactics, heuristic calculations, managed and unmanaged AI techniques, deep learning, and regularly animated calculations were some of the techniques used by the designers. Researchers have also used energy-harvesting techniques for smart grids, environmental systems, cities, and transportation systems as real data and benchmarks. What criteria are employed to assess IoT energy-harvesting models?

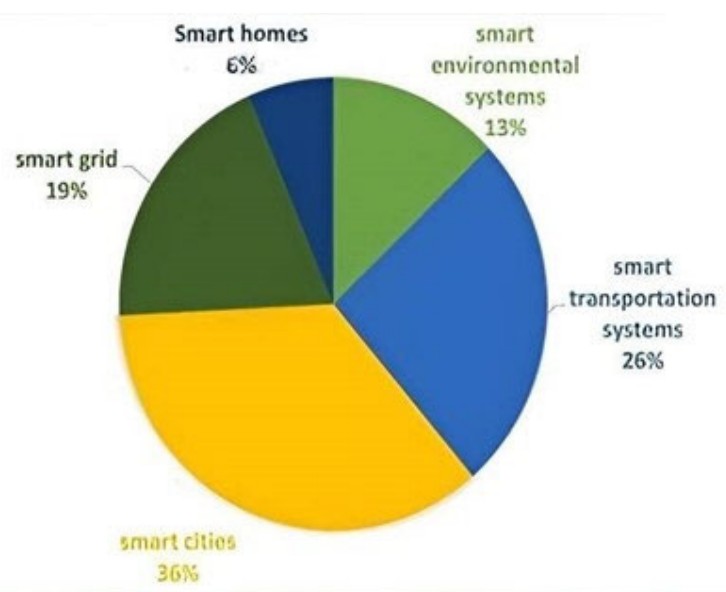

**Figure 9.** Variety of energy-harvesting management in the IoT.

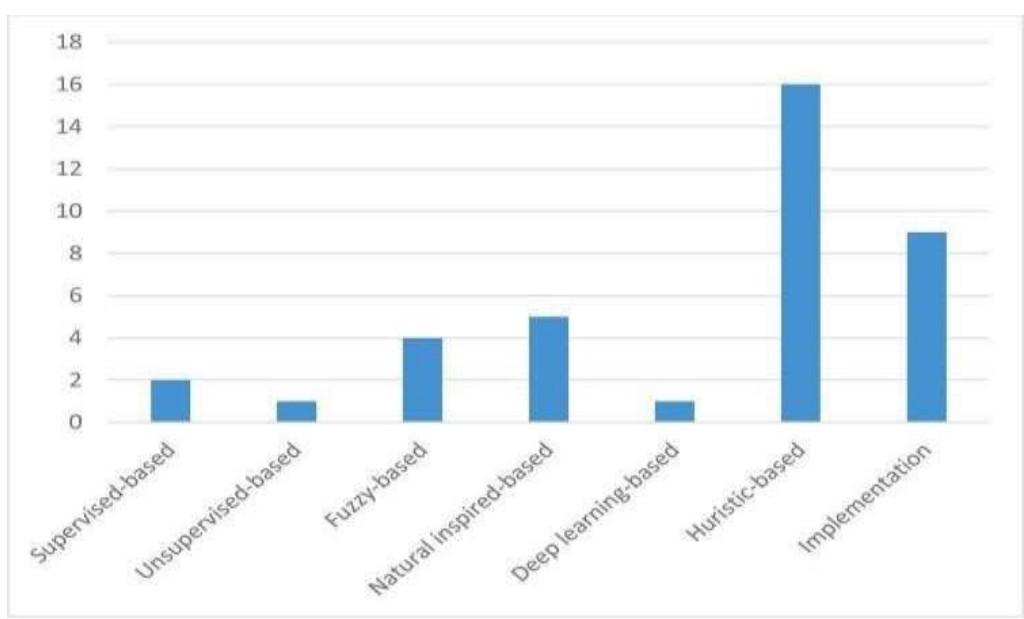

**Figure 10.** Percentage of energy-harvesting strategies for based on new innovations and algorithms.

Numerous measures, including energy consumption, throughput, response time, accuracy, precision, recall, F-score, and error rate, were used to study and evaluate the chosen research articles in light of Figure 11. In the conclusion, the experimental findings of the earlier investigations have taken accuracy, energy and power utilization, and response time into account. Additionally, throughput was given less consideration than the other elements. In addition to the eight crucial aspects we frequently outline, there are additional factors that should be taken into account, including security, performance effectiveness, and trust difficulties. What simulation tools and software have been considered for the Internet of Things' intelligent urban computing? Using MATLAB and the Java programming language in eclipse simulations, energy-harvesting approaches in intelligent urban computing have been created and tested [1,43,44] (Figure 12).

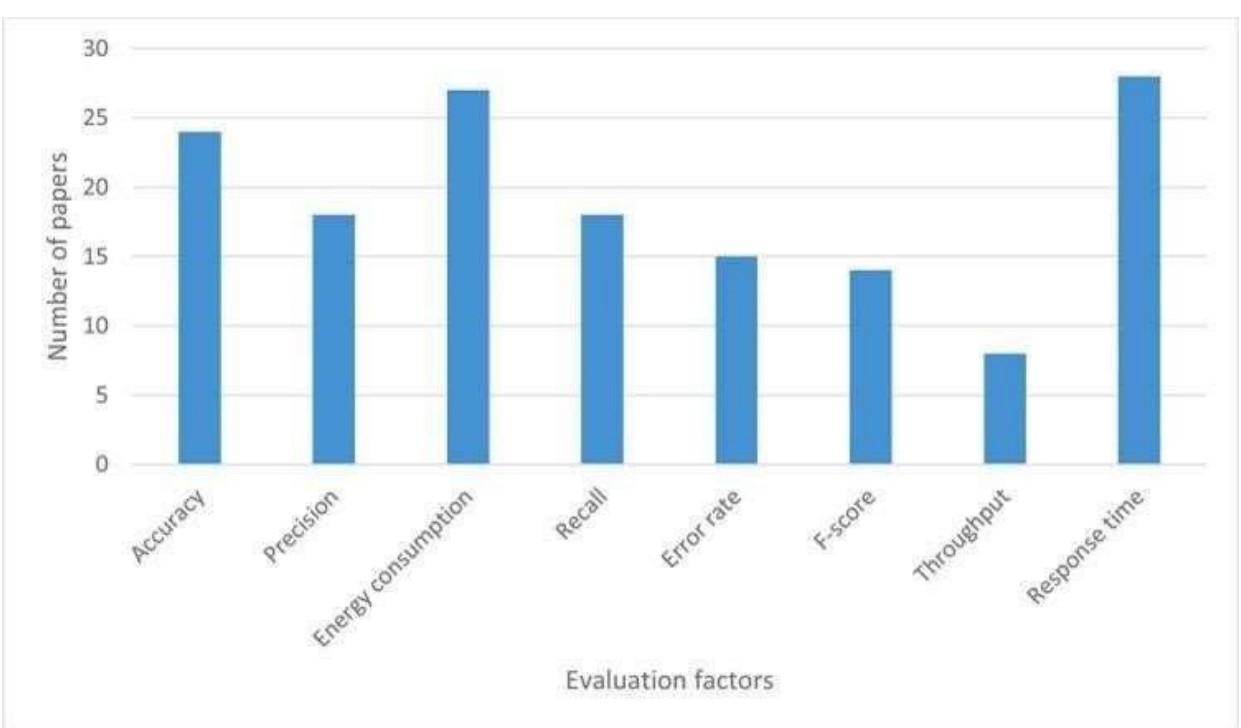

**Figure 11.** Comparison of evaluation factors in the energy-harvesting management approaches.

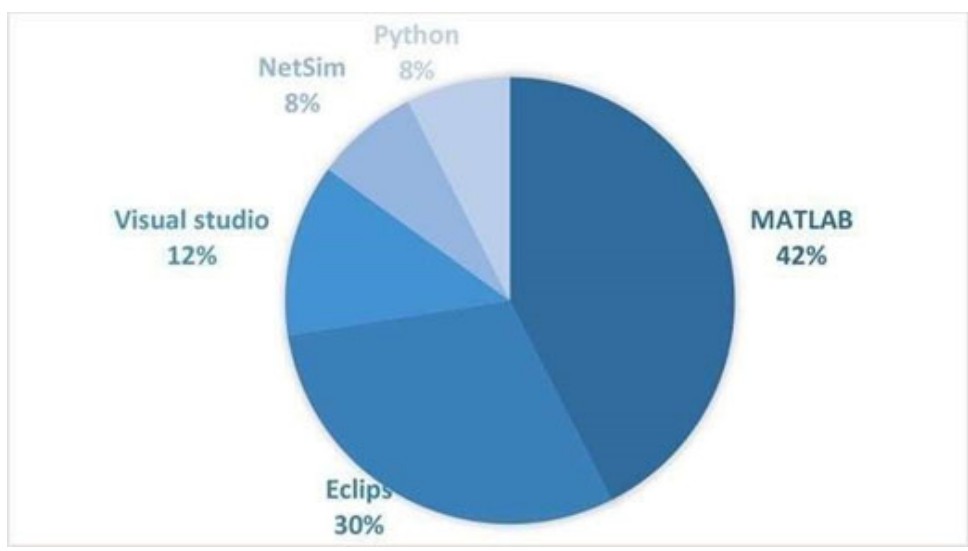

**Figure 12.** Comparison of simulation and implementation environments for this literature.

## 7. New Research Directions and Challenges

There are a few developments and study directions in the energy collection systems that have not been extensively investigated to open the door for future investigations in light of the aforementioned expert reports and results. Below is a brief explanation of several open issues. How to develop a substantial association between the energy consumption component and fault sites is one of the outstanding problems in fault prediction. Less failures can be supported by the smart industry and IoT by using federated learning and deep learning approaches [45]. Historical-based prediction is a problem that the smart sector needs to resolve in order to achieve the highest estimation accuracy. Formal estimating models are also viewed as new unanswered questions in order to demonstrate the accuracy of industrial maintenance.

The key unresolved issues in the energy-aware industrial IoT are test case time management and defect clustering. Furthermore, while evaluating test cases for industrial equipment, meta-heuristic testing approaches should be given more consideration in order to achieve the highest quality. Secure data-centric clustering, data filtering clustering, and extracting dependency matrix for evolutionary aspects of energy management strategies in sustainable smart cities are a few unresolved issues in energy-based healthcare architecture. In order to manage sensor life power consumption and save energy in mobile situations, the wireless sensor body area network is an essential part of applying energy-harvesting techniques. A multiobjective scenario based on energy use can also be used to evaluate the safety and security of these subjects. A critical tool for evaluating the precision and dependability of energy management procedures for smart home applications is conceptual formal approaches [46]. The reachability of the on/off power state, a deadlock-free position for renewing energy harvesting using solar panels, and exhibiting fairness requirements for locking doors and parking shelf doors are some of the most important aspects of this issue [47]. With the management of energy utilization in manufacturing and industrial equipment, big data analysis is now confronted with a new challenge [48]. Deep learning and machine learning are two examples of the new evolutionary algorithms that have been asserted to increase accuracy in industrial settings while consuming less energy.

### 7.1. Intelligent Energy Infrastructures for Future Smart Cities

The need to lessen the effects of climate change and global warming has led to an increase in the usage of renewable energies throughout the world. RE stands not only for a significant research and development industry, but also an efficient remedy considering the current economy in light of resource depletion and crucial to preserving the environment. More and more, AI is being used in this industry, improving results in terms of RE accessibility and efficiency. In a changing climate and market context, AI aids in managing energy production and consumption. One of the main issues facing this industry is the variability of renewable energy sources, which is becoming more of a problem as the proportion of RE in overall energy output rises. By performing predictive analysis, identifying patterns, lowering storage costs, and improving connectivity between grids and users, AI is the solution that should be implemented at the microeconomic level to increase grid stability, reliability, and sustainability.

The symbiotic relationship between RE and AI will transform the energy industry and advance sustainability on a national and international scale.

By identifying and forecasting patterns, completing specific activities without explicit human direction, streamlining the supply, and improving decision-making, AI could boost the efficiency of the RE sector. Due to quick forecasting and clever connections between key components brought about by the rapid development of AI-infused technology, it will offer superior insights into operations.

### 7.2. Determine the Cost and Time Requirements for Producing Sustainable Energy in Smart Cities

It can be difficult to quantify the time and money requirements for producing sustainable energy in smart cities because these requirements vary greatly depending on the particular objectives, scope, and conditions of each project. However, I can give you a broad overview of the variables affecting these demands and some approximations:

### 7.2.1. Financial Demands

1. Scale of the Project

USD 100,000 to USD 1 million for small-scale projects (such as a single building or neighborhood).

Medium-Scale Projects: USD 1 million to USD 10 million (ex., district-level renewable energy integration).

Large-Scale Projects: Depending on the size and energy requirements of the city, costs can range from USD 10 million to billions of dollars.

2. Technology Costs

Solar PV: installed kW prices range from USD 1000 to USD 4000;
Wind turbines: installed kW price range of USD 1200 to USD 2500;
Battery energy storage: USD 200 to USD 500 per kWh;
Upgrades to the grid: They can be substantial but vary widely;
Infrastructure and Grid Integration: Depending on the current status of the infrastructure, the costs for grid upgrades, smart grid technologies, and infrastructure enhancements may have a substantial impact on the budget.

3. Regulatory and Permitting Fees: Permit, Inspection, and Compliance Costs Can Vary, but They Make Up a Significant Portion of the Necessary Finances
4. Finance Charges: The Total Amount of Financial Requirements May Vary Depending on Loan Interest Rates, Financing Conditions, and Available Incentives

### 7.2.2. Time Requires

1. Studies of Planning and Feasibility: 2 Years to 6 Months
2. Permitting and Design: Depending on the Intricacy of the Project and the Necessary Regulations, 1 to 3 Years
3. Building and Installation

Larger-scale projects can take longer; medium-scale projects can take one to three years.

4. Grid Testing and Integration: 2 Years to 6 Months

Full Implementation: 3 to 10 years or longer, depending on the size and complexity of the project.

It is vital to keep in mind that these figures are only approximations and may differ greatly depending on the scope, location, technology, finance, and unanticipated difficulties of the project. The timeline and expenses may also be affected by developments in technology, modifications to regulations, and support or resistance from the community.

Continuous attempts to increase energy efficiency, encourage the use of renewable energy sources, and integrate smart grid technologies are common components of smart city programs. As a result, projects frequently have long-term sustainability objectives that go beyond the initial implementation stage.

It is advised to undertake a thorough feasibility study and collaborate with subject-matter experts who can provide accurate assessments based on the project's particular features and local conditions in order to obtain precise cost and time estimates for a specific sustainable energy project in a smart city.

### 8. Conclusions

The performance, efficacy, and accuracy of smart applications are significantly impacted by the employment of energy-harvesting technology to regulate energy savings and consumption in intelligent urban computing. The energy-harvesting management solutions for intelligent urban computing in the sustainable Internet of Things from 2019 to 2022 were examined in this article, which compiled 33 research publications. We discovered that the bulk of articles published on this subject occurred in the year 2019. The majority of papers—55% of the total—have been published in the IEEE magazine. Elsevier and Springer both own 14% and 25% of the total number of papers. We classified 33 carefully picked research articles into five sections to categorize energy management solutions, with smart transportation systems having the most significant influence on energy management and harvesting for Internet of Things (IoT) applications. Numerous sources of ambient energy are present in the human environment and could be investigated in order to progress the promising sectors of the Internet of Things and WSNs, particularly the IoT devices for the completion of intelligent cities. The typical energy sources that can be gathered are discussed in this essay. Energy can be conveniently gathered close to the application because it is readily available almost anyplace there are vibrations, daylight, heat, wind, radio frequency, water, or any other naturally occurring source. This will ensure that the EH

plans for the gleaming city enjoy all their benefits, including quick responses to numerous demands from different city zones and low maintenance costs. Energy-harvesting solutions have been suggested as a step toward green communication through the use of G-WPC and IoTs. In regard to edge-based intelligent urban computing's smart applications for sustainable and smart cities, this study looks into a range of green energy collection methods. Smart grids, smart environmental systems, smart transportation systems, and smart cities are the current four types of energy-harvesting techniques. This paper's technical review of energy-harvesting management strategies, including created algorithms, assessment standards, and evaluation environments, is intended to promote green urban computing.

**Author Contributions:** Conceptualization, R.S.; Formal analysis, R.S.; Investigation, R.S.; Resources, R.S.; Data curation, R.S.; Writing—original draft, R.S.; Supervision, F.A.-T. All authors have read and agreed to the published version of the manuscript.

**Funding:** This research received no external funding.

**Data Availability Statement:** Not applicable.

**Conflicts of Interest:** The authors declare no conflict of interest.

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
