# Peer review of "Sustainable Energy Production in Smart Cities"

_sustainability, doi:10.3390/su152216052_

Round 1
Reviewer 1 Report
The text needs to be more structured. It is too monolithic. In the introduction it is necessary to state the methodology of the work. There is also a lack of a subchapter Methodology/Methods of work. What is the added value of the article? What is the authors' research besides the fact that they collected 33 research articles and built on them? Where is the authors' own research in the article? What are the authors' suggestions? Quantify the financial and time demands of your suggestions.
Author Response
Response to Reviewer 1 Comments
Point 1: In the introduction it is necessary to state the methodology of the work. There is also a lack of a subchapter Methodology/Methods of work.
Response 1:
The literature review serves as the cornerstone of scientific writing since it allows the researcher to familiarize themselves with the texts and identify the most notable authors who have written on the subject. For the literature review, we used a systematic analysis technique. “Energy harvesting”, “Internet of Things”, and “sustainable smart cities” were the search phrases used in the systematic review to look for publications in large databases.
Point 2: What is the added value of the article? What is the authors' research besides the fact that they collected 33 research articles and built on them? Where is the authors' own research in the article?
Response 2:
This study investigates a variety of green energy collecting techniques in relation to edge-based intelligent urban computing's smart applications for sustainable and smart cities. The five categories of energy harvesting strategies currently in use are smart grids, smart environmental systems, smart transportation systems, and smart cities. The purpose of this paper is to provide a technical overview of energy harvesting management strategies for green urban computing, including developed algorithms, evaluation standards, and evaluation environments.
Point 3: What are the authors' suggestions?
Response 3:
- Discover a method to provide Internet of Things (IoT) nodes that have been placed with energy that is both widespread and long-lasting.
- This study has looked at the majority of urban sources for energy that have been described by scholars in the literature.
- Energy gathering has been proposed as a solution, enabling IoT hubs to hunt for self-sustaining energy from ecologically wide sources.
Point 4: Quantify the financial and time demands of your suggestions.
Response 4:
- List every task and the resources needed.
- Utilize Your Knowledge and Information.
- Observe the project budget.
- Determine the task's duration.

Reviewer 2 Report
Journal Sustainability (ISSN 2071-1050)
Manuscript ID sustainability-2298672
Type: Article
Number of Pages: 28
Title: Sustainable Energy Production in Smart Cities
Dear Authors,
It has been for me a great honour, as well as a pleasantly challenging activity, to review the article entitled “Sustainable Energy Production in Smart Cities.”
Overall, the article is interesting. It deals with vital issues related to sustainable energy production in smart cities (energy harvesting, internet of things etc.) which are important both from a scientific and practical point of view, which could make the study very utilitarian, however, there are some shortcomings in this regard, which I mention below with reference to the last chapter of the manuscript.
I think the paper has a good chance of attracting the attention of potential readers. However, I would suggest that the Authors introduce a few corrections (given below).
In my opinion, the Introduction chapter quite well introduces potential readers to the topics discussed by the Authors. However, it should be supported to a greater extent by the available and current literature on the subject. For example, since "Many definitions of this idea include the particular […]", please provide sources for such a statement, etc. In addition, the research gap in the existing literature on the subject should be outlined clearly and the novelties of this study should be indicated to a greater extent.
Chapters 2 – 5
These chapters are written in quite an interesting way, are divided into logically sequential sub-chapters. However, in my opinion, they are insufficiently supported by the existing and widely available literature on the subject. The whole work is based on only 20 literature references. Unfortunately, in my opinion, this is not sufficient, because this subject is currently widely discussed and described, and the Authors should clearly specify what knowledge is already available on this subject and what they bring to this area. In my opinion, it would also be worth referring to issues related to “smart energy” for the effective and efficient energy supply, which are currently widely discussed in the latest publications, e.g.:
https://doi.org/10.1016/j.energy.2017.05.123
https://doi.org/10.3390/en15228676
https://doi.org/10.3390/en15228418
6. Discussion
This chapter has potential and is the beginning of an interesting part, but too laconically conducted and again too poorly confronted with the available literature.
7. New research directions and challenges
I consider it valuable that the Authors indicate new research directions and challenges related to this.
The Conclusions are a bit to laconic. Furthermore, in my opinion, in this chapter it should be clearly defined who and how can benefit from the publication. This would determine the utilitarian nature of the work carried out. Besides, the Authors write that “This article collected 33 research papers”, but only 20 references are in the References section.
Summing up, in my opinion the topic of the article is interesting and has the potential to interest potential readers. I don't feel competent to comment on linguistic correctness as English is not my mother tongue.
I wish the Authors good luck.
Author Response
Response to Reviewer 2 Comments
Point 1: In my opinion, the Introduction chapter quite well introduces potential readers to the topics discussed by the Authors. However, it should be supported to a greater extent by the available and current literature on the subject. For example, since "Many definitions of this idea include the particular […]", please provide sources for such a statement, etc. In addition, the research gap in the existing literature on the subject should be outlined clearly and the novelties of this study should be indicated to a greater extent.
Response 1:
Smart cities are a cutting-edge concept for managing urban regions that will raise sustainability and quality of life for residents. Moreover, in order to improve ecological and economic sustainability, projects integrating digitalization and smart cities must generate value. However, by first deducting the rewards from the efforts, this increased value can be presented more plainly.
The challenge of building their infrastructure with modern methods that utilize little energy and have little impact on the environment is one that smart cities must face. Fighting climate change and other environmental problems requires the development of "smart buildings" and a more effective transportation system. A self-managing automated system that can transform electric power into a final product with little human involvement is necessary as part of a smart city's balanced energy exchange.
In order to balance power output and consumption, reduce generation capacity, and have an impact on other energy market participants, smart cities are developing a unified system that combines diverse energy, heat, gas, and water systems as well as telecommunications structures. The long-term health of the energy industry depends on electrification, the process of moving civilization toward using electricity as its primary energy source.
Point 2: Chapters 2 – 5 These chapters are written in quite an interesting way, are divided into logically sequential subchapters. However, in my opinion, they are insufficiently supported by the existing and widely available literature on the subject. The whole work is based on only 20 literature references. Unfortunately, in my opinion, this is not sufficient, because this subject is currently widely discussed and described, and the Authors should clearly specify what knowledge is already available on this subject and what they bring to this area. In my opinion, it would also be worth referring to issues related to “smart energy” for the effective and efficient energy supply, which are currently widely discussed in the latest publications, e.g.: https://doi.org/10.1016/j.energy.2017.05.123 https://doi.org/10.3390/en15228676 https://doi.org/10.3390/en15228418
Response 2:
Systems that use smart energy and smart energy
The names "Smart Energy" and "Smart Energy Systems" have been popular in recent years to describe a strategy that goes beyond what is meant by "Smart grid." Smart Energy Systems take a more integrated, holistic approach to the inclusion of more sectors (electricity, heating, cooling, industry, buildings, and transportation), as opposed to Smart Grids, which primarily focus on the electricity sector. This allows for the identification of more feasible and affordable solutions to the transformation into future renewable and sustainable energy solutions. This essay begins with a survey of the relevant scientific literature. The term "Smart Energy Systems" is then discussed in relation to the concerns of definition, solution identification, modeling, and storage integration. The concept of a smart energy system, it is concluded, represents a scientific paradigm shift away from single-sector thinking and toward an understanding of coherent energy systems that demonstrates how to profit from the fusion of all infrastructures and sectors.
A review of Polish urban development plans examining smart energy for smart cities
Understanding the current state and future advancements of smart energy techniques is crucial for the effective and efficient supply of energy to meet the needs of the exponentially increasing energy demands of modern cities. Smart energy is a vital component of the notion of a "Smart City." Using content analysis methods, this research examines how the Smart Energy agenda is included into Polish plans for developing smart cities. The most often mentioned elements of the Smart Energy agenda were the stakeholders' involvement, spatial aspects, Smart Energy conceptions, and Smart Energy key sectors. Universities, small enterprises, and public governance organizations are all essential key actors covered by stakeholders' involvement in Smart Energy agendas. The levels of the individual, city, regional (sub-regional), nation, and international (EU) are included in the spatial dimension components of the Smart Energy agenda, with the city level naturally dominating. The component for "Smart Energy conceptions" demonstrates a stark difference between "peripheral" Smart Energy conceptions and the four "core" Smart Energy conceptions (renewable energy, energy efficiency, energy-saving technology, and energy security). It was discovered that the only sectors mentioned in the analyzed urban development plans with relevance to the Smart Energy agenda were the building, transportation, lighting, and manufacturing sectors. The study's findings aid in a better knowledge of the Polish smart city and energy planning environments and may serve to enhance the cities' spatial planning tactics.
Taking into account energy costs and carbon emissions, stochastic operation optimization of the Smart Savona Campus as an Integrated Local Energy Community
Sector coupling increases the penetration of renewables and lowers carbon emissions while aiming to integrate various energy sectors and take use of synergies resulting from the interaction of various energy carriers. Sector coupling fits in well with the idea of an integrated local energy community (ILEC) at the local level. In an ILEC, active consumers make decisions about how to best meet their energy needs by coordinating the use of a variety of multi-carrier energy technologies. This results in greater economic and environmental advantages over the status quo. This article examines the stochastic operation optimization of the University of Genoa's smart Savona Campus in light of financial and environmental considerations. With two electrically connected multi-energy hubs that use absorption chillers, electric and geothermal heat pumps, solar thermal, combined heat and power systems, PV, and solar thermal, the campus is treated as an ILEC. With the right bidding tactics, the ILEC can take part in the day-ahead market (DAM) from this perspective. The fast forward selection algorithm is used to preserve the most representative scenarios while lessening the computational load of the following optimization phase. The roulette wheel method is used to generate an initial set of scenarios for solar irradiance. In order to optimize the operation strategies of the various technologies in the ILEC as well as the bidding strategies of the ILECs in the DAM, both energy costs and carbon emissions are taken into account through a multi-objective approach. This is accomplished through the use of mixed-integer linear programming (MILP). Results from a case study demonstrate how the ILEC's ideal bidding techniques on the DAM enable minimization of the users' net daily cost. Similarly to environmental optimization, the ILEC also runs in self-consumption mode.
Point 3: 6. Discussion
This chapter has potential and is the beginning of an interesting part, but too laconically conducted and again too poorly confronted with the available literature.
Response 3:
We use technical reports to respond to these enquiries in accordance with the prepared questions in Section 3 as follows: What intelligent urban computing models for the Internet of Things are being investigated in terms of management tactics for energy harvesting? Figure 9 illustrates how much energy harvesting has already been done for Internet of Things models of intelligent urban computing. Eight futuristic transportation options There are four studies each on smart home and environmental systems, as well as six studies on the smart grid. Smart houses and smart environmental systems each have four research articles, suggesting that both techniques still face two unresolved problems even though smart cities have a higher impact on the management of energy harvesting for intelligent urban computing in IoT. To lower the energy and power consumption of IoT devices, established metropolitan areas are studying and implementing smart solutions for smart city environments. RQ2: What optimization strategies and models are investigated in this literature? In light of Fig. 10, the heuristic-based technique is used more frequently than the alternatives. The developers used heuristic calculations to evaluate how effectively smart homes, smart cities, and smart transportation systems gathered energy in accordance with the specified scientific classification. Studies on smart transportation have examined energy management using fuzzy methodologies based on fuzzy logic models. The authors evaluated the current energy-aware solutions in case studies, including smart cities, using fuzzy, heuristic, and natural-inspired algorithms. For the management of smart grids, authors have used supervised machine learning methods, heuristic algorithms, and strategies derived from nature. Then, a list of useful optimization techniques is created for efficient environmental management. Fluffy tactics, heuristic calculations, managed and unmanaged AI techniques, deep learning, and regularly animated calculations were some of the techniques used by the designers. Researchers have also used energy harvesting techniques for smart grids, environmental systems, cities, and transportation systems as real data and benchmarks. What criteria are employed to assess IoT energy harvesting models?
Numerous measures, including as energy consumption, throughput, response time, accuracy, precision, recall, F-score, and error rate, were used to study and evaluate the chosen research articles in light of Fig. 11. In the conclusion, the experimental findings of the earlier investigations have taken accuracy, energy and power utilization, and response time into account. Additionally, throughput was given less consideration than the other elements. In addition to the eight crucial aspects we frequently outline, there are additional factors that should be taken into account, including security, performance effectiveness, and trust difficulties. What simulation tools and software have been considered for the Internet of Things' intelligent urban computing? Using MATLAB and the Java programming language in eclipse simulations, energy harvesting approaches in intelligent urban computing have been created and tested[31–33].
Point 4: 7. New research directions and challenges
I consider it valuable that the Authors indicate new research directions and challenges related to this.
Response 4:
Intelligent Energy Infrastructures for Future Smart Cities
The need to lessen the effects of climate change and global warming has led to an increase in the usage of renewable energies throughout the world. RE stands for more than just a significant research and development industry, but also an efficient remedy Considering the current economy in light of resource depletion and crucial to preserving the environment. More and more, AI being used into this industry, improving results in terms of RE accessibility and efficiency. In a changing climate and market context, AI aids in managing energy production and consumption. One of the main issues facing this industry is the variability of renewable energy sources, which is becoming more of a problem as the proportion of RE in overall energy output rises. By performing predictive analysis, identifying patterns, lowering storage costs, and improving connectivity between grids and users, AI is the solution that should be implemented at the microeconomic level to increase grid stability, reliability, and sustainability.
The symbiotic relationship between RE and AI will transform the energy industry and advance sustainability on a national and international scale.
By identifying and forecasting patterns, completing specific activities without explicit human direction, streamlining the supply, and improving decision-making, AI could boost the efficiency of the RE sector. Due to quick forecasting and clever connections between key components brought about by the rapid development of AI-infused technology, it will offer superior insights into operations.
Point 5: The Conclusions are a bit to laconic. Furthermore, in my opinion, in this chapter it should be clearly defined who and how can benefit from the publication. This would determine the utilitarian nature of the work carried out.
Response 5:
The performance, efficacy, and accuracy of smart applications are significantly impacted by the employment of energy harvesting technology to regulate energy savings and consumption in intelligent urban computing. The energy harvesting management solutions for intelligent urban computing in the sustainable Internet of Things from 2019 to 2022 were examined in this article, which compiled 33 research publications. We discovered that the bulk of articles published on this subject occurred in the year 2019. The majority of papers—55% of the total—have been published in the IEEE magazine. Elsevier and Springer both own 14% and 25% of the total number of papers. We classified 33 carefully picked research articles into five sections to categorize energy management solutions, with smart transportation systems having the biggest influence on energy management and harvesting for Internet of Things (IoT) applications. Numerous sources of ambient energy are present in the human environment and could be investigated in order to progress the promising sectors of the Internet of Things and WSNs, particularly the IoT devices for the completion of intelligent cities. The typical energy sources that can be gathered are discussed in this essay. Energy can be conveniently gathered close to the application because it is readily available almost anyplace there is vibration, daylight, heat, wind, radio frequency, water, or any other naturally occurring source. This will ensure that the EH plans for the gleaming city enjoy all of their benefits, including quick responses to numerous demands from different city zones and low maintenance costs. Energy harvesting solutions have been suggested as a step toward green communication through the use of G-WPC and IoTs. In regard to edge-based intelligent urban computing's smart applications for sustainable and smart cities, this study looks into a range of green energy collection methods. Smart grids, smart environmental systems, smart transportation systems, and smart cities are the current five types of energy harvesting techniques. This paper's technical review of energy harvesting management strategies, including created algorithms, assessment standards, and evaluation environments, is intended to promote green urban computing.

Reviewer 3 Report
sustainability-2298672
Sustainable Energy Production in Smart Cities
It seems that this is a review article. After reviewing the paper thoroughly, I found a lack of information in the proposed work. Some points need to be known.
- It will be good to add a table showing different types of sensors used for smart cities.
- Also, add a table that includes different IoT networks used in smart cities e.g. LoRa, WsNs, Bluetooth etc., showing their advantages and disadvantages, energy consumption etc.
- It is better to list a comparison table to compare results with previous work.
- References are too low for a review paper. Please include more (latest) references. e.g., Khan, A.U.; Khan, M.E.; Hasan, M.; Zakri, W.; Alhazmi, W.; Islam, T. An Efficient Wireless Sensor Network Based on the ESP-MESH Protocol for Indoor and Outdoor Air Quality Monitoring. Sustainability 2022, 14, 16630. https://doi.org/10.3390/su142416630
Author Response
Response to Reviewer 3 Comments
Point 1: It will be good to add a table showing different types of sensors used for smart cities.
Resonse 1:
By 2050, 85% of the world's population, according to experts, would reside in urban areas. Cities should therefore be ready to meet the needs of its residents and offer the greatest services. The smart city, a more effective system that maximizes its resources and services via the use of monitoring and communication technologies, is a typical representation of the idea of a future metropolis. Making the shift to smart cities is thus one of the measures that cities all around the world can take to become more sustainable. Here, sensors are crucial to the system because they collect pertinent data from the city, its residents, and the accompanying communication networks that transmit it in real time. Although there are many applications for these sensors, they can be divided into six categories: energy, health, mobility, security, water, and waste management. This review includes an overview of several sensors that are frequently utilized in initiatives to create smart cities based on these groupings. There are insights regarding various applications and communication technologies as well as the primary potential and difficulties encountered when converting to a smart city. In the end, this process is about more than just smart urban infrastructure; it's also about how these new digitalization and sensing advancements enhance quality of life. Smarter communities are those that invest in, socialize with, and adapt to these technologies in accordance with local and regional societal requirements and ideals. Privacy and disruptions to cyber security continue to be major vulnerabilities.
Table:1 “Types of sensors utilized in various smart city application divisions.”
|
Subcategory |
Sensing Parameters |
Type of Sensors |
Distance GW-Sensor |
|
|
Agriculture |
Humidity, Temperature, Luminosity, Solar radiation, Soil, Conductivity, Ph |
Ultrasonic Temperature Humidity Soil |
30 cm–15 km |
|
|
Healthcare |
Health signs |
Biosensors |
3 m |
|
|
Energy |
Light intensity Motion Voltage Temperature Humidity |
Temperature Humidity Motion |
15 km |
|
|
Traffic |
Motion Occupancy |
Magnetic Ultrasonic |
500 m–1 km |
|
|
Environment |
CO2, NO2, O3 Concentration Weather |
Gas Temperature |
200 cm–5 km |
|
Point 2: Also, add a table that includes different IoT networks used in smart cities e.g. LoRa, WsNs, Bluetooth etc., showing their advantages and disadvantages, energy consumption etc.
Resonse 2:
Implemented IoT Technologies in Smart Cities:
The technologies used to build smart cities are listed in Table 2 along with their general application and component breakdown. The applications of smart cities, both current and future, integrate all the technology already in use.
Table 2 Implemented IoT Technologies in Smart Cities
|
Technology |
Main components used |
Purpose |
|
LPWANs (Low Power Wide Area Networks) |
- |
Connecting All Type Of Sensors In A Single Network |
|
RFID (Radio Frequency Identification) |
Tags and Readers |
Transmitting And Receiving Data |
|
WSNs (Wireless Sensor Networks) |
- |
Detecting Various Environmental Factors |
|
Li-Fi (Light Fidelity) |
LEDs |
High Speed Wireless Internet Communication |
|
MQTT (Message Queuing Telemetry Transport) |
- |
Translates The Messages Between Devices, Servers And Applications |
Point 3:
References are too low for a review paper. Please include more (latest) references. e.g., Khan, A.U.; Khan, M.E.; Hasan, M.; Zakri, W.; Alhazmi, W.; Islam, T. An Efficient Wireless Sensor Network Based on the ESP-MESH Protocol for Indoor and Outdoor Air Quality Monitoring. Sustainability 2022, 14, 16630. https://doi.org/10.3390/su142416630
Response 4:
[31] Khan, A. U., Khan, M. E., Hasan, M., Zakri, W., Alhazmi, W., & Islam, T. (2022). An Efficient Wireless Sensor Network Based on the ESP-MESH Protocol for Indoor and Outdoor Air Quality Monitoring. Sustainability, 14(24), 16630.

Round 2
Reviewer 1 Report
Please, reply carefully my original question: "Quantify the financial and time demands of your suggestions" and add the answer to the conclusion.
Author Response
Response to Reviewer 1 Comments - (Round 2)
Point 1: Please, reply carefully my original question: "Quantify the financial and time demands of your suggestions" and add the answer to the conclusion.
Resonse 1:
It can be difficult to quantify the time and money requirements for producing sustainable energy in smart cities because these requirements vary greatly depending on the particular objectives, scope, and conditions of each project. However, I can give you a broad overview of the variables affecting these demands and some approximations:
7.2.1 Financial Demands:
7.2.1.1 Scale of the Project:
$100,000 to $1 million for small-scale projects (such as a single building or neighborhood).
Medium-Scale Projects: $1 million to $10 million (ex., district-level renewable energy integration).
Large-Scale Projects: Depending on the size and energy requirements of the city, costs can range from $10 million to billions of dollars.
7.2.1.2 Technology Costs:
Solar PV: installed kW prices range from $1,000 to $4,000.
Wind turbines: installed kW price range of $1,200 to $2,500.
Battery energy storage: $200 to $500 per kWh.
Upgrades to the grid: They can be substantial but vary widely.
Infrastructure and Grid Integration: Depending on the current status of the infrastructure, the costs for grid upgrades, smart grid technologies, and infrastructure enhancements may have a substantial impact on the budget.
7.2.1.3 Regulatory and Permitting Fees: Permit, inspection, and compliance costs can vary, but they make up a significant portion of the necessary finances.
7.2.1.4 Finance Charges: The total amount of financial requirements may vary depending on loan interest rates, financing conditions, and available incentives.
7.2.2 Time Requires:
7.2.2.1 Studies of Planning and Feasibility: 2 years to 6 months.
7.2.2.2 Permitting & Design: Depending on the intricacy of the project and the necessary regulations, 1 to 3 years.
7.2.2.3 Building and Installation:
Larger-scale projects can take longer; medium-scale projects can take one to three years.
7.2.2.4 Grid testing and integration: 2 years to 6 months.
Full Implementation: 3 to 10 years or longer, depending on the size and complexity of the project.
It's vital to keep in mind that these figures are only approximations and may differ greatly depending on the scope, location, technology, finance, and unanticipated difficulties of the project. The timeline and expenses may also be affected by developments in technology, modifications to regulations, and support or resistance from the community.
Continuous attempts to increase energy efficiency, encourage the use of renewable energy sources, and integrate smart grid technologies are common components of smart city programs. As a result, projects frequently have long-term sustainability objectives that go beyond the initial implementation stage.
It is advised to undertake a thorough feasibility study and collaborate with subject-matter experts who can provide accurate assessments based on the project's particular features and local conditions in order to obtain precise cost and time estimates for a specific sustainable energy project in a smart city.

Reviewer 2 Report
In my opinion, the Authors made satisfactory corrections and provided clear answers.
Author Response
Response to Reviewer 2 Comments - (Round 2)
Point 1: In my opinion, the Authors made satisfactory corrections and provided clear answers.
Response 1:
I appreciate you, reviewer

Reviewer 3 Report
sustainability-2298672
Title: Sustainable Energy Production in Smart Cities
Thank you for allowing me to revise resubmitted manuscript titled "Sustainable Energy Production in Smart Cities" I believe the submitted manuscript and presented work is suitable for publishing in Sensors except for some minor revision.
Minor revision:
-Abstract should be precise.
-Quality of all the figures shoud be improved.
-The query “add a table that includes different IoT networks used in smart cities e.g. LoRa, WsNs, Bluetooth etc., showing their advantages and disadvantages, energy consumption etc.” is not adequately addressed. Please read the comments carefully.
Author Response
Response to Reviewer 3 Comments - (Round 2)
Point 1: Abstract should be precise.
Resonse 1:
Finding a method to provide the installed Internet of Things (IoT) nodes with energy that is both ubiquitous and long-lasting is crucial for assuring continuous smart city optimization. These and other problems have impeded new research into energy harvesting. After the COVID-19 pandemic and the lockdown that all but ended daily activity in many countries, the ability of human remote connection to enforce social distance became crucial. Since they lay the groundwork for surviving a lockdown, Internet of Things (IoT) devices are once again widely recognised as crucial elements of the smart city. The recommended solution of energy collection would enable IoT hubs to search for self-sustaining energy from ecologically large sources. The bulk of urban energy sources that could be used have been looked at in this work, according to descriptions made by researchers in the literature. Given the abundance of free resources in the city covered in this research, we have also suggested that energy sources can be application-specific. This implies that energy needs for various IoT devices or wireless sensor networks (WSNs) for smart city automation should be scavenged near to those needs. One of the important smart ecological and energy-harvesting subjects that has evolved as a result of the advancement of intelligent urban computing is intelligent cities and societies. Collecting and exchanging Internet of Things (IoT) gadgets and smart applications that improve people's quality of life is the main goal of a sustainable smart city. Energy harvesting management, a key element of sustainable urban computing, is hampered by the exponential rise of Internet of Things (IoT) sensors, smart apps, and complicated populations. These challenges include the requirement to lower the associated elements of energy consumption, power conservation, and waste management for the environment. However, the idea of energy harvesting management for sustainable urban computing is currently expanding at an exponential rate and requires attention due to regulatory and economic constraints. This study investigates a variety of green energy collecting techniques in relation to edge-based intelligent urban computing's smart applications for sustainable and smart cities. The five categories of energy harvesting strategies currently in use are smart grids, smart environmental systems, smart transportation systems, and smart cities. In terms of developed algorithms, evaluation criteria, and evaluation environments, this review's objective is to discuss the technical features of energy harvesting management systems for environmentally friendly urban computing. For sustainable smart cities, which specifically contribute to increasing the energy consumption of smart applications and human life in complex and metropolitan areas, it is crucial from a technical perspective to develop existing barri-ers and unexplored research trajectories in energy harvesting and waste management.
Point 2: Quality of all the figures should be improved
Resonse 2:
All figures quality have been improved
Point 3:
Add a table that includes different IoT networks used in smart cities e.g. LoRa, WsNs, Bluetooth etc., showing their advantages and disadvantages, energy consumption etc.”
Response 3:
IoT networks used in smart cities, such as LoRa, Bluetooth, Wireless Sensor Networks (WSNs), and others, are listed in the following table, along with details on their advantages, disadvantages, and energy consumption:
|
IoT Network |
Advantages |
Disadvantages |
Energy Usage |
|
LoRa |
- Long-range communication |
- Low data rate |
- Low power |
|
- Low power consumption |
- Limited bandwidth |
||
|
- Scalability |
- Not suitable for real-time applications |
||
|
WSNs |
- Scalability |
- Limited range |
- Variable, depends on node activity |
|
- Low power consumption |
- Network topology maintenance |
||
|
- Suitable for sensor data collection |
- Limited processing capabilities |
||
|
Bluetooth |
- Low power consumption |
- Short range |
- Low power |
|
- Wide device compatibility |
- Interference in crowded environments |
||
|
- Low cost |
- Limited scalability |
||
|
Zigbee |
- Low power consumption |
- Limited range |
- Low power |
|
- Scalability |
- Interference from other wireless devices |
||
|
- Suitable for home automation |
- Limited data rate |
||
|
NB-IoT |
- Wide-area coverage |
- Costly infrastructure deployment |
- Low power |
|
- Low power consumption |
- Limited bandwidth |
||
|
- Suitable for large-scale deployments |
- Relatively higher device cost |
||
|
5G |
- High data rates |
- Infrastructure deployment cost |
- Variable, depends on usage |
|
- Low latency |
- Limited range |
||
|
- Supports massive IoT devices |
- Energy-intensive for small devices |
|
Note that this table gives a broad overview of the advantages, disadvantages, and energy consumption of several IoT networks. Depending on the implementation, hardware selection, and deployment conditions, the actual performance and energy use may differ. Planning IoT deployments in smart cities requires careful analysis based on the particular needs of your project in order to select the best network technology.
